# A genetic variant of fatty acid amide hydrolase (FAAH) exacerbates hormone-mediated orexigenic feeding in mice

Georgia Balsevich[1]*, Gavin N Petrie[1], Daniel E Heinz[2], Arashdeep Singh[3], Robert J Aukema[1], Avery C Hunker[4], Haley A Vecchiarelli[1], Hiulan Yau[1], Martin Sticht[1], Roger J Thompson[1], Francis S Lee[5], Larry S Zweifel[4], Prasanth K Chelikani[6], Nils C Gassen[2], Matthew N Hill[1]*

[1]Hotchkiss Brain Institute, University of Calgary, Calgary, Canada; [2]Neurohomeostasis Research Group, Department of Psychiatry and Psychotherapy, University Hospital Bonn, Bonn, Germany; [3]Monell Chemical Senses Center and Department of Neuroscience, University of Pennsylvania, Philadelphia, United States; [4]Department of Psychiatry and Behavioral Sciences, University of Washington, Seattle, United States; [5]Weill Cornell Medical College, Cornell University, New York, United States; [6]Texas Tech University, School of Veterinary Medicine, Amarillo, United States

*For correspondence:
georgia.balsevich@ucalgary.ca
(GB);
mnhill@ucalgary.ca (MNH)

**Competing interest:** The authors declare that no competing interests exist.

**Abstract** Fatty acid amide hydrolase (FAAH) degrades the endocannabinoid anandamide. A polymorphism in *FAAH* (FAAH C385A) reduces FAAH expression, increases anandamide levels, and increases the risk of obesity. Nevertheless, some studies have found no association between FAAH C385A and obesity. We investigated whether the environmental context governs the impact of FAAH C385A on metabolic outcomes. Using a C385A knock-in mouse model, we found that FAAH A/A mice are more susceptible to glucocorticoid-induced hyperphagia, weight gain, and activation of hypothalamic AMP-activated protein kinase (AMPK). AMPK inhibition occluded the amplified hyperphagic response to glucocorticoids in FAAH A/A mice. FAAH knockdown exclusively in agouti-related protein (AgRP) neurons mimicked the exaggerated feeding response of FAAH A/A mice to glucocorticoids. FAAH A/A mice likewise presented exaggerated orexigenic responses to ghrelin, while FAAH knockdown in AgRP neurons blunted leptin anorectic responses. Together, the FAAH A/A genotype amplifies orexigenic responses and decreases anorexigenic responses, providing a putative mechanism explaining the diverging human findings.

## Editor's evaluation

This study work provides evidence that an enzyme in key neurons in the brain regulate body weight. They used novel mouse models to mimic mutations in this gene in humans. The work is significant as it reconciles previously contradictory clinical data. Thus, the studies will be of wide interest.

## Introduction

Since 1975, the prevalence of worldwide obesity has nearly tripled and has reached epidemic proportions (*World Health Organization, 2020*). An incomplete understanding surrounding the etiology of obesity has hindered the development of therapeutic interventions. Regardless, it is well known that both genetic and environmental factors contribute to its development (*Goodarzi, 2018*). Therefore, it is important to identify how specific genetic and environmental factors interact to contribute to the mechanisms underlying the etiology of obesity.

The endocannabinoid system (ECS), comprised of two receptors (CB1 and CB2), endogenous cannabinoids (*N*-arachidonoylethanolamide, [anandamide], AEA and 2-arachidonoylglycerol), and their synthetic and degradative enzymes, is recognized as a powerful regulator of energy homeostasis and body weight (*Mazier et al., 2015*). In general, activation of the ECS promotes energy storage, food intake, and increased weight gain. Therefore, components of the ECS are perfectly positioned to moderate individual susceptibility to obesity. In humans, there is a common genetic variant in the gene for fatty acid amide hydrolase (FAAH) (*Sipe et al., 2002*), the primary enzyme responsible for the inactivation of the endocannabinoid AEA. This missense mutation in FAAH (C385A; rs324420) destabilizes the FAAH protein, reduces AEA metabolism, and increases AEA signaling (*Chiang et al., 2004*; *Sipe et al., 2002*). Importantly, increased body mass index (BMI) has been associated with the low-expressing FAAH variant (A-allele carriers) (*Durand et al., 2008*; *Monteleone et al., 2008*; *Sipe et al., 2005*; *Zhang et al., 2009*). This agrees with the well-known role of AEA to stimulate feeding and enhance energy storage through the activation of CB1 receptor signaling (*Mazier et al., 2015*). However, the human data surrounding the C385A variant of FAAH and body weight regulation are oftentimes conflicting, with some studies showing no effect of FAAH C385A on metabolic outcomes (*Jensen et al., 2007*; *Lieb et al., 2009*). The conflicting findings may be attributed to differences in an individual's environmental context and resulting endocrine milieu. In this regard, gene × environment interactions could govern whether the FAAH C385A genotype affects obesity susceptibility under particular environmental contexts.

Environmental conditions dictate an individual's physiological endocrine state. For example, chronic stress and high fat diet feeding are associated with shifts in levels of circulating hormones that regulate appetite and energy homeostasis, including, but not limited to, glucocorticoids (GCs), ghrelin, and leptin (*Bouassida et al., 2010*; *Schwarz et al., 2011*; *Ulrich-Lai and Ryan, 2014*). Interestingly, the metabolic actions of orexigenic signals, such as GCs and ghrelin, increase endocannabinoid signaling (*Balsevich et al., 2017*; *Edwards and Abizaid, 2016*). By contrast, the anorectic peptide leptin suppresses endocannabinoid signaling to mediate its effects (*Balsevich et al., 2018*; *Di Marzo et al., 2001*). Therefore, differences in FAAH expression, due to distinct genetic FAAH variants, may influence the metabolic outcomes resulting from shifts in endocrine states. Specifically, we have previously shown that chronic exposure to GCs leads to obesity through an endocannabinoid-mediated mechanism (*Bowles et al., 2015*). It was likewise shown that GCs increase the activity of AMP-activated protein kinase (AMPK), through a CB1 receptor-dependent mechanism in the hypothalamus to promote weight gain (*Scerif et al., 2013*). Indeed hypothalamic AMPK plays an integral role in the regulation of feeding and body weight, where its activation leads to increased feeding (*Andersson et al., 2004*; *Kim et al., 2004*; *Minokoshi et al., 2004*). Similar to GCs, the orexigenic gastric hormone ghrelin increases the activity of hypothalamic AMPK and consequently feeding through an CB1 receptor-dependent mechanism (*Kola et al., 2008*). Taken together, we hypothesized that the discrepancies in the human literature for FAAH C385A and body weight outcomes are attributed to differences in their environmental context and ultimately their endocrine milieu. Therefore, we sought to characterize the behavioral, cellular, and molecular changes associated with the FAAH C385A variant in response to GC (corticosterone [CORT]) exposure using the FAAH C385A knock-in mouse model, recapitulating the common human mutation in the *FAAH* gene. We furthermore assessed a role for FAAH in ghrelin- and leptin-dependent feeding outcomes to determine the generalizability of these effects.

## Results

### Body weight and body composition are similar between FAAH C/C and FAAH A/A mice under basal conditions

As a first step, we examined the metabolic outcomes arising in FAAH C/C (wild-type [WT]) and FAAH A/A (HOM) mice under baseline conditions. FAAH C/C and FAAH A/A mice presented similar body weights, fat mass, and lean mass at baseline (*Figure 1*, *Figure 1—figure supplement 1*). Yet despite no difference in body weight or composition, there was a genotype × time trend (p=0.082) for 24 hr cumulative food intake, whereby post hoc analysis indicated that FAAH A/A mice present decreased food intake compared to FAAH C/C mice. In support of this finding, 24 hr energy expenditure was lower in FAAH A/A mice compared to FAAH C/C mice. Furthermore, FAAH A/A mice presented an

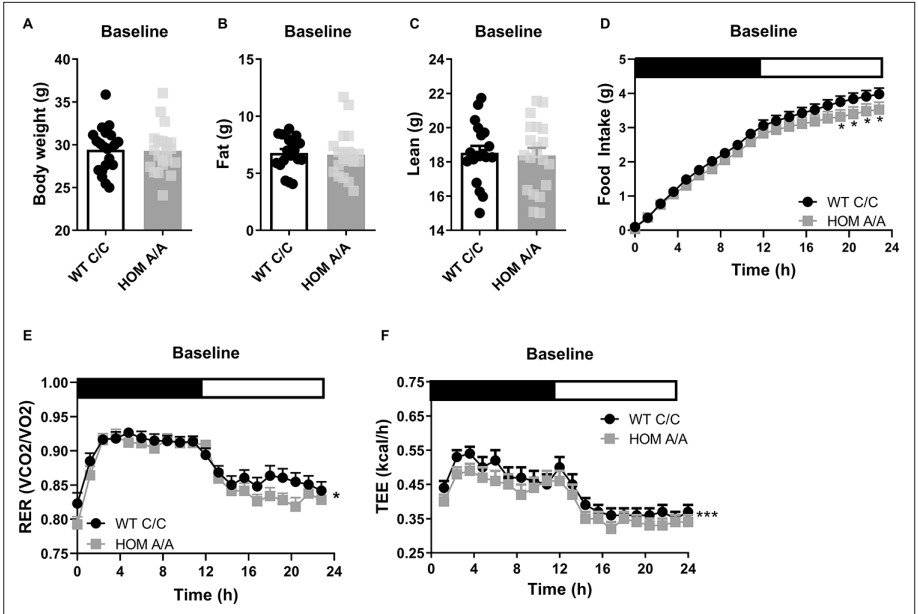

**Figure 1.** The fatty acid amide hydrolase (FAAH) C385A variant has no effect on baseline body weight or body composition. (**A–C**) Under basal conditions, there is no effect of the FAAH C385A variant on body weight (T(39) = 0.1516, p=0.88), fat mass (T(39) = 0.2640, p=0.79), or lean mass (T(39) = 0.4219, p=0.6754). (**D–F**) By contrast, the C385A variant modifies underlying metabolic parameters, wherein FAAH A/A (HOM) mice present reduced food intake (genotype × time: F(76, 2360.107)=1.238, p=0.082 plus post hoc testing), decreased respiratory exchange ratio (RER) (genotype: F(1,197.189)=6.406, p=0.012), and decreased energy expenditure (genotype: F(1, 387.547)=22.949, p<0.001) compared to FAAH C/C (wild-type [WT]) mice. For panels (**A–C**), n=20 WT; n=21 HOM. For panels (**D–F**), n=17 per genotype. Data are presented as means ± SEM, and in Panel (**F**) estimated means adjusted for body weight are presented. Panels (**A–C**) were analyzed by unpaired Student's t-tests. Panels (**D–F**) were analyzed by linear mixed models with repeated measured. Asterisks denote significant genotype effect. *p<0.05, ***p<0.001.

The online version of this article includes the following source data and figure supplement(s) for figure 1:

**Source data 1.** Basal Body Weight of FAAH C/C and FAAH A/A mice across ages.

**Figure supplement 1.** Basal Body Weight of FAAH C/C and FAAH A/A mice across ages.

overall lower respiratory exchange ratio (RER), favoring fat utilization compared to FAAH C/C mice. Collectively, these data suggest that under baseline conditions, there is no effect of the C385A variant on body weight or composition despite differences in underlying metabolic parameters. The lower food intake in FAAH A/A mice matches their lower energy expenditure. These findings support the human studies reporting no association between A-allele carriers and obesity susceptibility under standard feeding conditions.

## FAAH A/A mice are more sensitive to the prolonged effects of GCs

To examine whether the FAAH C385A variant affects GC-mediated metabolic outcomes, we characterized the metabolic phenotypes arising in FAAH C/C (WT) and FAAH A/A (HOM) mice following 4-week exposure to CORT (25 µg/ml) or vehicle (1% ethanol) delivered through their drinking water as previously described (*Karatsoreos et al., 2010*). At baseline, circulating CORT was comparable between FAAH C/C and FAAH A/A mice (*Figure 2—figure supplement 1*). Similarly, prolonged CORT exposure significantly elevated circulating CORT measured during the dark (active) stage regardless of genotype. There was furthermore a corresponding decrease in adrenal weights following prolonged CORT treatment, supporting previous reports (*Karatsoreos et al., 2010*). With regards to metabolic outcomes, prolonged CORT exposure led to significant body weight gain and increased fat mass regardless of genotype (*Figure 2*). Furthermore, a genotype × treatment trend (p=0.0520) followed by post hoc analysis indicated that FAAH A/A mice gained significantly more weight on account of CORT treatment compared to FAAH C/C mice. There was neither an effect of genotype nor treatment

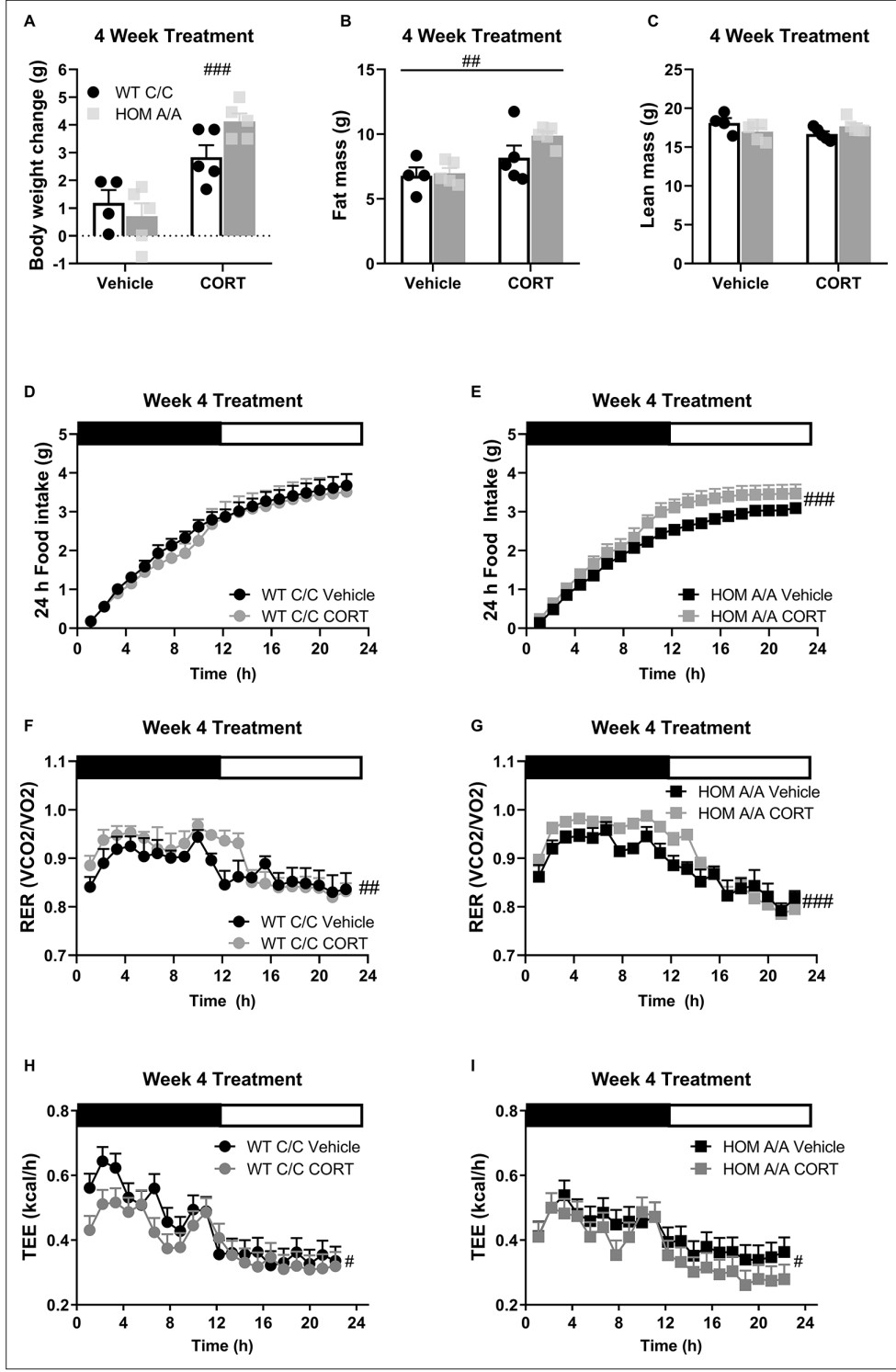

**Figure 2.** Fatty acid amide hydrolase (FAAH) A/A mice show exaggerated body weight gain and food intake on account of prolonged corticosterone (CORT). (**A**) Body weight gain was increased following 4 weeks of CORT treatment (25 µg/ml in drinking water) (treatment $F_{(1,15)}$ = 6.65, p<0.001), where only CORT-treated HOM A/A mice gained significantly more weight than their vehicle-treated counterparts. CORT-treated wild-type (WT) C/C mice did not gain significantly more weight than vehicle-treated WT C/C mice (treatment × genotype = $F_{(1,15)}$=4.456, p=0.052 plus post hoc testing). (**B**) Fat mass was increased from 4 weeks of CORT exposure independent of genotype (treatment $F_{(1, 15)}$=11.66, p=0.004). (**C**) A significant interaction was detected for lean mass (treatment × genotype: $F_{(1, 15)}$=5.111, p=0.0391) but Bonferroni's post hoc testing revealed no

*Figure 2 continued on next page*

*Figure 2 continued*

effect of either genotype or prolonged CORT exposure. (**D–E**) Whereas 4 weeks of CORT treatment did not affect cumulative food intake measured over 24 hr in FAAH C/C mice (treatment: $F_{(1, 20.758)} = 0.820$, $p = 0.320$), it significantly increased food intake in FAAH A/A mice (treatment: $F_{(1, 24.770)} = 24.538$, $p < 0.001$). (**F–G**) Regardless of genotype, 4 weeks of CORT exposure increased respiratory exchange ratio (RER) (treatment: WT ($F_{(1, 47.659)} = 4.289$, $p = 0.044$); HOM $F_{(1, 68.057)} = 19.493$, $p < 0.001$), (**H–I**) whereas decreased energy expenditure measure over 24 hr (treatment: WT $F_{(1, 47.659)} = 4.289$, $p = 0.044$; HOM $F_{(1, 68.057)} = 19.493$, $p < 0.001$). For panels (**A–I**), n=4 WT-vehicle; n=5 WT-CORT, n=5 HOM-vehicle; n=5 HOM-CORT. Data are presented as means ± SEM, and in Panels (**H** and **I**) estimated means adjusted for body weight are presented. Panels (**A–C**) were analyzed by two-way ANOVAs. Panels (**D–I**) were analyzed by linear mixed models with repeated measures. Pound signs denote significant treatment effect. Asterisks denote significant genotype effect. *$p < 0.05$, #$p < 0.05$, ##$p < 0.01$, ###$p < 0.001$.

The online version of this article includes the following source data and figure supplement(s) for figure 2:

**Source data 1.** Endocrine measures in FAAH C385A mice at baseline and under Vehicle and CORT conditions.

**Figure supplement 1.** Endocrine measures in FAAH C385A mice at baseline and under Vehicle and CORT conditions.

**Figure supplement 2.** Feeding throughout the 4-week treatment regime in FAAH C/C and FAAH A/A mice.

on lean mass. Importantly, prolonged CORT treatment significantly increased 24 hr cumulative food intake exclusively in FAAH A/A mice (***Figure 2***). In accordance with this, food intake throughout the 4-week treatment was significantly elevated by CORT solely in FAAH A/A mice (***Figure 2—figure supplement 2***). However, regardless of genotype, prolonged CORT exposure favored carbohydrate utilization as reflected in the increased RER across genotypes as well as decreased total energy expenditure (TEE) (***Figure 2***). Therefore, FAAH A/A mice present a heightened sensitivity to GC-mediated hyperphagia and weight gain following prolonged exposure.

## FAAH A/A mice are more sensitive to the immediate effects of GCs

It is well known that short- and long-term exposures to GCs have distinct outcomes on the ECS (reviewed by ***Balsevich et al., 2017***). Therefore, we investigated whether the FAAH C385A variant likewise modifies the metabolic responses to CORT exposure within the initial 48 hr of treatment onset. Similar to 30-day CORT exposure, FAAH A/A mice present heightened sensitivity to the immediate effects of GC exposure on food intake and weight gain. Specifically, within 48 hr of treatment onset, food intake was increased on account of oral CORT exposure exclusively in FAAH A/A mice (***Figure 3***). Likewise, within the first 24 hr of treatment onset, CORT exposure led to weight gain exclusively in FAAH A/A mice. However, regardless of genotype, RER was elevated on account of 48 hr CORT exposure (***Figure 3—figure supplement 1***). Furthermore, TEE was exclusively decreased in FAAH C/C mice on account of CORT exposure. Indeed CORT exposure is known to decrease energy expenditure (***Hardwick et al., 1989***; ***Ramage et al., 2016***; ***Strack et al., 1995***; ***van den Beukel et al., 2014***). The lack of CORT-induced suppression of energy expenditure in FAAH A/A mice is consistent with their observed hyperphagic response to CORT, which is known to elevate energy expenditure on account of diet-induced thermogenesis (***Westerterp, 2017***), thus possibly countering the CORT-mediated decrease in energy expenditure. Finally, we employed implanted telemeters to examine total activity under basal conditions and during the initial 24 hr of CORT exposure. Neither CORT treatment nor genotype influenced home-cage activity (***Figure 3—figure supplement 2***).

Given our findings that the FAAH C385A variant specifically modifies GC-mediated feeding effects, we next examined AEA content in brain regions known to play an integral role in endocannabinoid-mediated feeding (***Ruiz de Azua and Lutz, 2019***) to determine whether CORT exposure differentially mobilizes AEA in FAAH C/C and FAAH A/A mice. Remarkably, in the hypothalamus, AEA content was exclusively elevated in FAAH A/A mice on account of 48 hr CORT (***Figure 3***). By contrast, in the ventral tegmental area and nucleus accumbens, there was no significant effect of either CORT treatment or genotype (***Figure 3—figure supplement 3***). Furthermore, there was no effect of the FAAH C385A polymorphism on AEA levels in peripheral organ systems that are recognized as the primary mediators of endocannabinoid-mediated regulation of metabolism (***Osei-Hyiaman et al., 2008***; ***Pagotto et al., 2006***; ***Ruiz de Azua et al., 2017***), namely brown adipose tissue, inguinal white adipose tissue, epididymal WAT, and liver. However, in the liver, 48 hr CORT significantly elevated AEA levels independent of genotype. Taken together, FAAH A/A mice are more sensitive to CORT-induced hyperphagia and

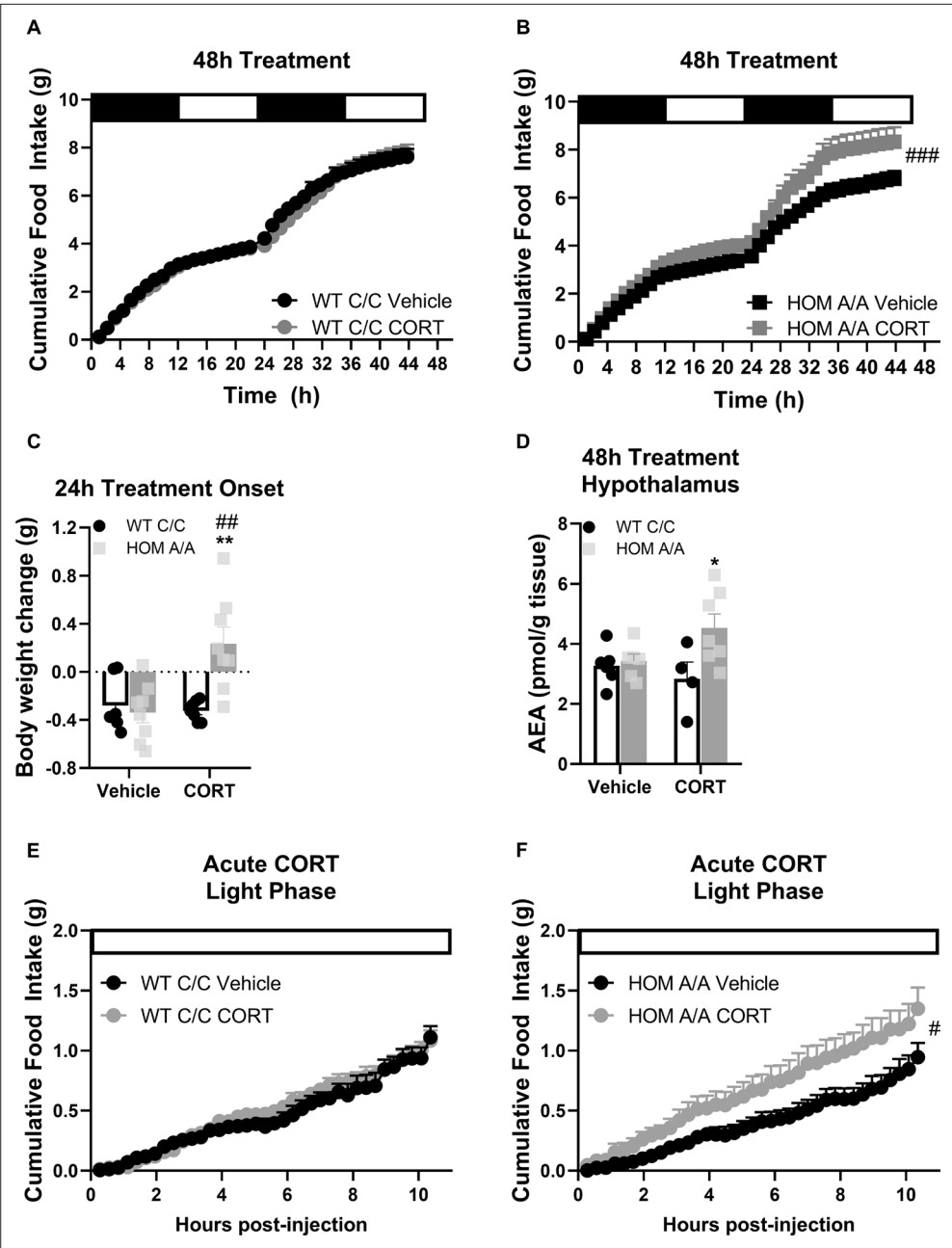

**Figure 3.** Fatty acid amide hydrolase (FAAH) A/A mice are hyperresponsive to the immediate orexigenic effects of corticosterone (CORT). (**A–B**) Within the first 48hr of treatment onset, CORT exposure (25 µg/ml in drinking water) had no effect on cumulative food intake in FAAH C/C (wild-type [WT]) mice (treatment: $F_{(1, 57.970)}=0.600$, p=0.442) whereas it significantly increased food intake in FAAH A/A (HOM) mice (treatment: $F_{(1, 55.289)}=17.202$, p<0.001). (**C**) Within 24 hr of treatment onset, CORT exposure led to weight gain exclusively in FAAH A/A mice (treatment × genotype: $F_{(1, 26)}=10.06$, p=0.004 plus Bonferroni's post hoc testing). (**D**) Likewise, hypothalamic AEA levels were increased from 48 hr CORT treatment exclusively in FAAH A/A mice (treatment × genotype: $F_{(1, 20)}=4.177$, p=0.054 plus Bonferroni's post hoc testing). (**E–F**) Acute CORT (3 mg/kg, single i.p. injection) again had no effect on cumulative food intake in FAAH C/C mice (treatment: $F_{(1, 20.236)}=0.482$, p=0.495) yet increased light phase food intake in FAAH A/A mice (within the initial 10 hr following CORT injections) (treatment: $F_{(1, 14.720)}=4.729$, p=0.046). For Panel (A): n=9 per group. For Panel (B), n=10 HOM-vehicle; n=11 HOM-CORT. For Panel (C), n=7 WT-vehicle; n=7 WT-CORT; n=8 HOM-vehicle; n=8 HOM_CORT. For Panel (D), n=7 WT-vehicle; n=4 WT-CORT; n=6 HOM-vehicle; n=7 HOM-CORT. For Panel (E) n=8 per group. For Panel (F): n=8 HOM-vehicle; n=7 HOM-CORT. Data are presented as means ± SEM. Panels (**A–B**) and (**E–F**) were analyzed by linear mixed

*Figure 3 continued on next page*

*Figure 3 continued*

models with repeated measures. Panels (**C–D**) were analyzed by two-way ANOVAs. Pound signs denote significant treatment effect. Asterisks denote significant genotype effect. *p<0.05, **p<0.01, #p<0.05, ##p<0.01, ###p<0.001.

The online version of this article includes the following source data and figure supplement(s) for figure 3:

**Source data 1.** Substrate utilization and energy expenditure throughout the 48h treatment regime in FAAH C385A mice.

**Figure supplement 1.** Substrate utilization and energy expenditure throughout the 48h treatment regime in FAAH C385A mice.

**Figure supplement 2.** Home-cage activity at baseline and throughout 24h CORT treatment in FAAH C/C and FAAH A/A mice.

**Figure supplement 3.** Anandamide (AEA) levels measured in distinct brain regions and peripheral tissues of FAAH C385A mice after 48h Vehicle or CORT treatment.

**Figure supplement 4.** The lasting effects of acute CORT on metabolic outcomes in FAAH C/C and FAAH A/A mice.

weight gain following 48 hr exposure, which is accompanied by increased hypothalamic AEA levels exclusively in CORT-treated FAAH A/A mice.

GCs are known to signal through genomic and non-genomic signaling mechanisms to elicit both delayed and rapid behavioral effects, respectively (*Balsevich et al., 2017*). In order to decipher the signaling mechanism through which the FAAH C385A variant impacts GC-mediated feeding outcomes, we examined the feeding responses of FAAH C/C and FAAH A/A mice to an acute (single) injection of CORT (3 mg/kg). CORT was delivered at the onset of the light/inactive phase, a period of low feeding, to capture any hyperphagic effects of CORT. In agreement with our previous findings, an acute injection of CORT significantly elevated light phase feeding exclusively in the FAAH A/A mice within 10 hr (*Figure 3*). The effects of CORT on cumulative food intake were no longer evident by 24 hr post-injection (*Figure 3—figure supplement 4*). Accordingly, there was no effect of acute CORT treatment on body weight gain examined 24 hr post-injection. In terms of TEE and RER, acute CORT injections significantly reduced energy expenditure whereas promoted carbohydrate utilization regardless of genotype (*Figure 3—figure supplement 4*). Our data indicate that the FAAH C385A variant selectively exaggerates the hyperphagic effects of GCs.

## Hypothalamic AMPK signaling is downstream of AEA-GC-mediated feeding effects

As the effect of the FAAH C385A variant on GC-dependent outcomes was exclusive to feeding, we next addressed the underlying mechanism responsible for mediating this interaction. Given a previous study demonstrating that GCs increase hypothalamic AMPK activity through an endocannabinoid-dependent mechanism (*Scerif et al., 2013*), we examined the effects of the FAAH C385A variant on GC-mediated effects on hypothalamic AMPK signaling. We first examined the phosphorylation status of AMPK and the direct targets of AMPK as a proxy of AMPK signaling activation. For this purpose, FAAH C/C and A/A mice were injected with CORT (3 mg/kg) or vehicle, and exactly 1 hr later, hypothalamic tissue was collected. For protein quantification, we used capillary-based immunoblotting (BioTechne, ProteinSimple, WES), a novel technique that allows automated separation and detection of proteins from lysates with very low amounts. We found that AMPK signaling was markedly increased exclusively in FAAH A/A mice following administration of CORT. Specifically, while we did not detect a significant increase in AMPK activity per se (as reflected through the phosphorylation status of AMPK), we observed a significant increase in the activation status of direct downstream AMPK targets (as reflected through the increased phosphorylation status of targets expressing the AMPK-consensus sequence) in CORT-treated FAAH A/A mice (*Figure 4*). To complement these findings, we treated immortalized hypothalamic cells (GT1-7 cells) with the FAAH inhibitor URB597 to mimic the increased AEA signaling present in FAAH A/A mice. Following 4 hr pretreatment with either URB597 (1 µM) or vehicle, GT1-7 cells were treated for an additional 2 hr with CORT (1 µM) or vehicle in a 2×2 design. In support of our previous findings, CORT treatment significantly increased AMPK signaling activity exclusively in URB597-treated cells as reflected in increased AMPK phosphorylation and increased phosphorylation status of downstream targets of AMPK (*Figure 4*).

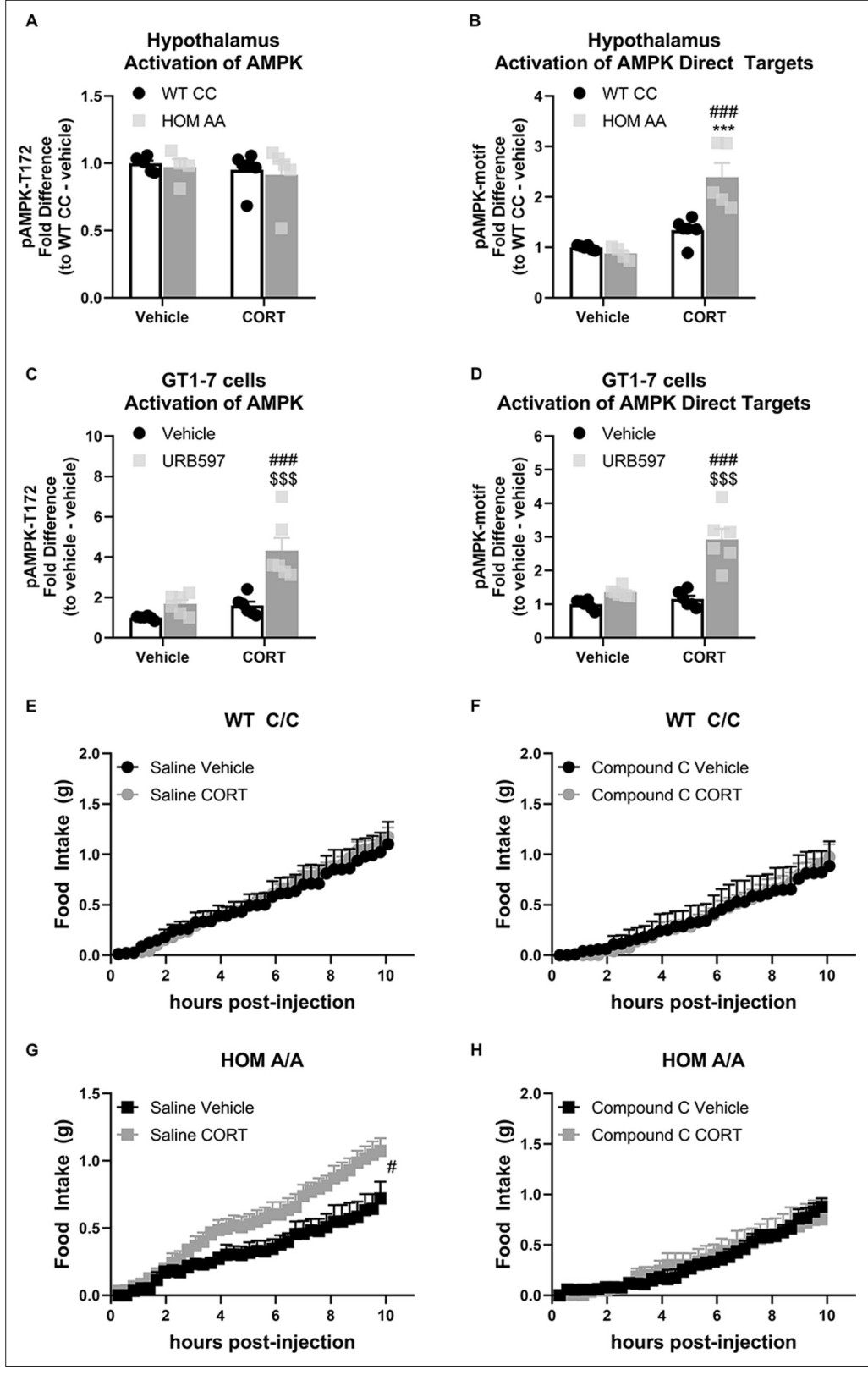

**Figure 4.** The fatty acid amide hydrolase (FAAH) A/A genotype amplifies the glucocorticoid (GC)-mediated orexigenic effects through an AMP-activated protein kinase (AMPK)-dependent mechanism. (**A**) Hypothalamic AMPK activity (as assessed by pAMPK protein expression) was unaffected by genotype ($F_{(1, 17)}=0.249$, $p=0.624$) or corticosterone (CORT) (3 mg/kg, i.p.) ($F_{(1, 17)}=0.711$, $p=0.411$). (**B**) However, in the hypothalamus, downstream

*Figure 4 continued on next page*

*Figure 4 continued*

targets of AMPK were activated from CORT exposure exclusively in FAAH A/A (HOM) mice ($F_{(1, 17)}=15.45$, p=0.001 plus post hoc testing). (**C–D**) In GT1-7 hypothalamic cells, AMPK activity was enhanced exclusively in CORT-treated cells pretreated with URB597, assessed by pAMPK ($F_{(1, 20)}=8.807$, p=0.008 plus post hoc testing) and pAMPK-motif ($F_{(1, 20)}=16.74$, p<0.001 plus post hoc testing) protein expression. (**E–F**). Acute CORT treatment had no effect on cumulative light phase food intake in FAAH C/C (wild-type [WT]) mice following administration of either saline (treatment: $F_{(1, 11.155)}=0.45$, p=0.836) or an AMPK inhibitor (compound C, 5 mg/kg, i.p.) (treatment: $F_{(1, 8.755)}=0.001$, p=0.982). By contrast, (**G–H**) AMPK inhibition prevented CORT-mediated orexigenic effects in FAAH A/A mice measured during the light phase (treatment: saline ($F_{(1, 14.344)}=6.112$, p=0.027); compound C ($F_{(1, 8.793)}=0.009$, p=0.927)). Panels (**A–B**): n=6 WT-vehicle biological replicates; n=6 WT-CORT biological replicates; n=4 HOM-vehicle biological replicates; n=5 HOM-CORT biological replicates. Panels (**C–D**): n=6 biological replicates per group. Panel (**E**): n=4 WT-vehicle; n=8 WT-CORT. Panel (**F**): n=4 HOM-vehicle; n=6 HOM-CORT. Panel (**G**): n=6 WT-vehicle. N=7 WT-CORT. Panel (**H**): n=5 per group. Protein expression data are expressed as relative fold change compared to control condition ± SEM. Food intake data are presented as means ± SEM. Panels (**A–D**) were analyzed by two-way ANOVAs. Panels (**E–H**) were analyzed by linear mixed models with repeated measures. Pound signs denote significant treatment effect. Asterisks denote significant genotype effect. Dollar signs denote significant URB597 treatment effect. ***p<0.001, #p<0.05, ##p<0.001, $$$p<0.001.

The online version of this article includes the following source data and figure supplement(s) for figure 4:

**Source data 1.** Lasting effects of acute CORT treatment following AMPK inhibition in FAAH C/C and A/A mice on feeding.

**Figure supplement 1.** Lasting effects of acute CORT treatment following AMPK inhibition in FAAH C/C and A/A mice on feeding.

To assess the functional implications of these FAAH-dependent events, we examined the effects of AMPK inhibition on CORT-mediated feeding outcomes in the FAAH C/C and A/A mice. At 1 hr prior to CORT/vehicle injections, FAAH C/C and A/A mice were treated with compound C (5 mg/kg, i.p.), an AMPK inhibitor, or saline. As expected, FAAH C/C mice did not show a feeding response to CORT (during the light phase) nor was there a CORT effect in the FAAH C/C mice when AMPK was inhibited (*Figure 4*). By contrast, FAAH A/A mice showed the expected hyperphagic response to CORT treatment during their light phase feeding, which was blocked by pretreatment with the AMPK inhibitor. After 24 hr, feeding returned to baseline (*Figure 4—figure supplement 1*). Taken together, our data demonstrates that the FAAH C385A variant amplifies the effects of CORT on feeding through an AMPK-dependent mechanism.

## FAAH knockdown exclusively in hypothalamic AgRP neurons is sufficient to mediate the exaggerated hyperphagic response to GCs and ablate the anorexigenic effects of leptin

Within the hypothalamus, different neuronal populations regulate feeding. For example, activation of agouti-related protein (AgRP) neurons is known to drive feeding (*Krashes et al., 2011*). Interestingly, it has been shown that GCs increase the activity of AgRP neurons to elicit feeding (*Perry et al., 2019*). Therefore, we sought to assess the impact of knocking down FAAH expression exclusively in AgRP neurons on CORT-mediated feeding responses. Here, we used a cre-dependent CRISPR/SaCas9 AAV-based system containing an HA-tagged SaCas9 with an sgRNA directed against FAAH (SaCas9-FAAH) or control (SaCas9-Control) vector (*Hunker et al., 2020*). To validate these constructs, we examined FAAH activity in *CaMKIIα-cre* mice following intra-hippocampal delivery of the AAV vectors. CaMKIIα drives cre expression in the forebrain, predominantly in CA1 pyramidal cells of the hippocampus (*Tsien et al., 1996*), where FAAH is highly expressed (*Egertová et al., 1998*; *Gulyas et al., 2004*; *Tsou et al., 1998*). Using this strategy, we are able to assess changes in FAAH function on account of CRISPR/SaCas9 gene editing in the hippocampus of *CaMKIIα-cre* mice, where it is otherwise unfeasible in *AgRP-Ires-cre* mice on account of relatively low FAAH activity in the sparsely distributed AgRP neurons. We found that mutagenesis of FAAH (using SaCas9-FAAH) significantly decreased FAAH activity relative to the controls (using SaCas9-Control) in *CaMKIIα-cre* mice (*Figure 5—figure supplement 1*). To directly examine the role of FAAH in AgRP neurons on CORT-mediated feeding, these AAV-CRISPR/SaCas9 constructs were stereotaxically injected into the arcuate nucleus of the hypothalamus of *Agrp-Ires-cre* mice to facilitate mutagenesis exclusively in AgRP neurons. HA-tagged

protein expression in the arcuate nucleus was probed to confirm cre-mediated DNA recombination. The HA-tag was detected exclusively in the ARC, in a pattern reminiscent of AgRP neurons (*Figure 5A*) and in a pattern similar to previously published expression profiles in Agrp-ires cre mice using cre-dependent viral delivery systems (*Krashes et al., 2011*; *Krashes et al., 2014*). Furthermore, no expression was detected in non-cre-expressing mice (*Figure 5A*).

Following a minimum of 4 weeks' recovery to allow for viral expression and gene knockout, we compared feeding responses to CORT in control mice (AgRP$^{control}$) and in mice where FAAH had been knocked down in AgRP neurons (AgRP$^{FAAH}$). Body weights were similar between AgRP$^{control}$ and AgRP$^{FAAH}$ mice at the onset of CORT treatment (*Figure 5B*). In AgRP$^{control}$ mice, there was no hyperphagic effect from a single CORT (3 mg/kg) injection (*Figure 5C*). However, and consistent with GC's effects in FAAH A/A mice, CORT significantly increased light phase feeding in AgRP$^{FAAH}$ mice (*Figure 5D*). Interestingly, we found lower basal food intake in AgRP$^{FAAH}$ mice relative to AgRP$^{control}$ mice under vehicle conditions, mimicking the phenotype observed in FAAH A/A mice under basal conditions (*Figure 1*). Taken together, suppression of FAAH exclusively in hypothalamic AgRP neurons recapitulates the exaggerated hyperphagic response to CORT observed in FAAH A/A mice.

We have previously shown that FAAH A/A mice are unresponsive to the anorectic effects of acute leptin exposure (*Balsevich et al., 2018*), which agrees with the literature indicating that leptin suppresses endocannabinoid signaling to mediate its feeding responses (*Balsevich et al., 2018*; *Di Marzo et al., 2001*). We therefore investigated whether FAAH knockdown exclusively in AgRP neurons is also sufficient to blunt leptin feeding responses. For this purpose, we allowed AgRP$^{control}$ and AgRP$^{FAAH}$ mice to recover for 5 weeks following the single CORT injection. Following recovery, AgRP$^{control}$ and AgRP$^{FAAH}$ mice were fasted overnight and then injected with either vehicle (saline) or leptin (2 mg/kg) immediately before receiving food. As expected, leptin reduced body weight gain and food intake following an overnight fast in AgRP$^{control}$ mice (*Figure 5E–F*). However, AgRP$^{FAAH}$ mice were insensitive to leptin's effects on weight gain or feeding (*Figure 5G–H*), indicating that FAAH knockdown exclusively in hypothalamic AgRP neuron is sufficient to block the anorexigenic effects of leptin. Similar to the effects of CORT, this demonstrates that knockdown of FAAH explicitly in AgRP neurons is sufficient to recapitulate the altered effects of leptin seen in the FAAH C385A mice (*Balsevich et al., 2018*).

## FAAH A/A mice are more sensitive to the acute hyperphagic effects of ghrelin

Endocannabinoids not only play a central role in GC-mediated metabolic outcomes, but furthermore are integral to the orexigenic effects of ghrelin (*Kola et al., 2008*; *Tucci et al., 2004*). Therefore, we further investigated the generalizability of this phenomenon to other orexigenic signals beyond GC, and as such explored whether the FAAH C385A variant affects ghrelin-mediated feeding effects. To address this question, we examined the feeding responses of FAAH C/C and FAAH A/A mice to a single, subthreshold injection of ghrelin (1 mg/kg) or saline delivered during the light/inactive phase to capture the hyperphagic effects of ghrelin during a period of low feeding. Similar to acute GC exposure, a single exposure to ghrelin increased light phase feeding exclusively in FAAH A/A mice (*Figure 6*). By 24 hr post-injection, the effects of ghrelin on cumulative food intake were no longer evident (*Figure 6—figure supplement 1*). There was also no lasting effect of ghrelin on body weight gain measured at 24 hr post-injection. Ghrelin likewise significantly increased RER during the light phase exclusively in FAAH A/A mice, which agrees with the hyperphagic effect (*Figure 6—figure supplement 1*). The elevated RER tended (p=0.068) to continue 24 hr post-ghrelin injections in FAAH A/A mice. By contrast, there was no effect of ghrelin on energy expenditure in either FAAH C/C or A/A mice.

Similar to hypothalamic GC signaling, ghrelin also activates AMPK signaling in the hypothalamus to increase feeding through an endocannabinoid-dependent manner (*Kola et al., 2008*). To determine whether hypothalamic AMPK is differentially affected by the FAAH C385A variant, we again examined the phosphorylation status of AMPK and the direct targets of AMPK as a proxy of AMPK signaling activation in FAAH C/C and A/A mice. Ghrelin (1 mg/kg) significantly increased hypothalamic levels of phosphorylated AMPK and direct downstream AMPK targets specifically and exclusively in FAAH A/A mice 1 hr post-injection (*Figure 6*). Together, our data indicate that the effects of the FAAH C385A variant are not selective for GC-mediated effects. Rather, the FAAH C385A variant exaggerates both

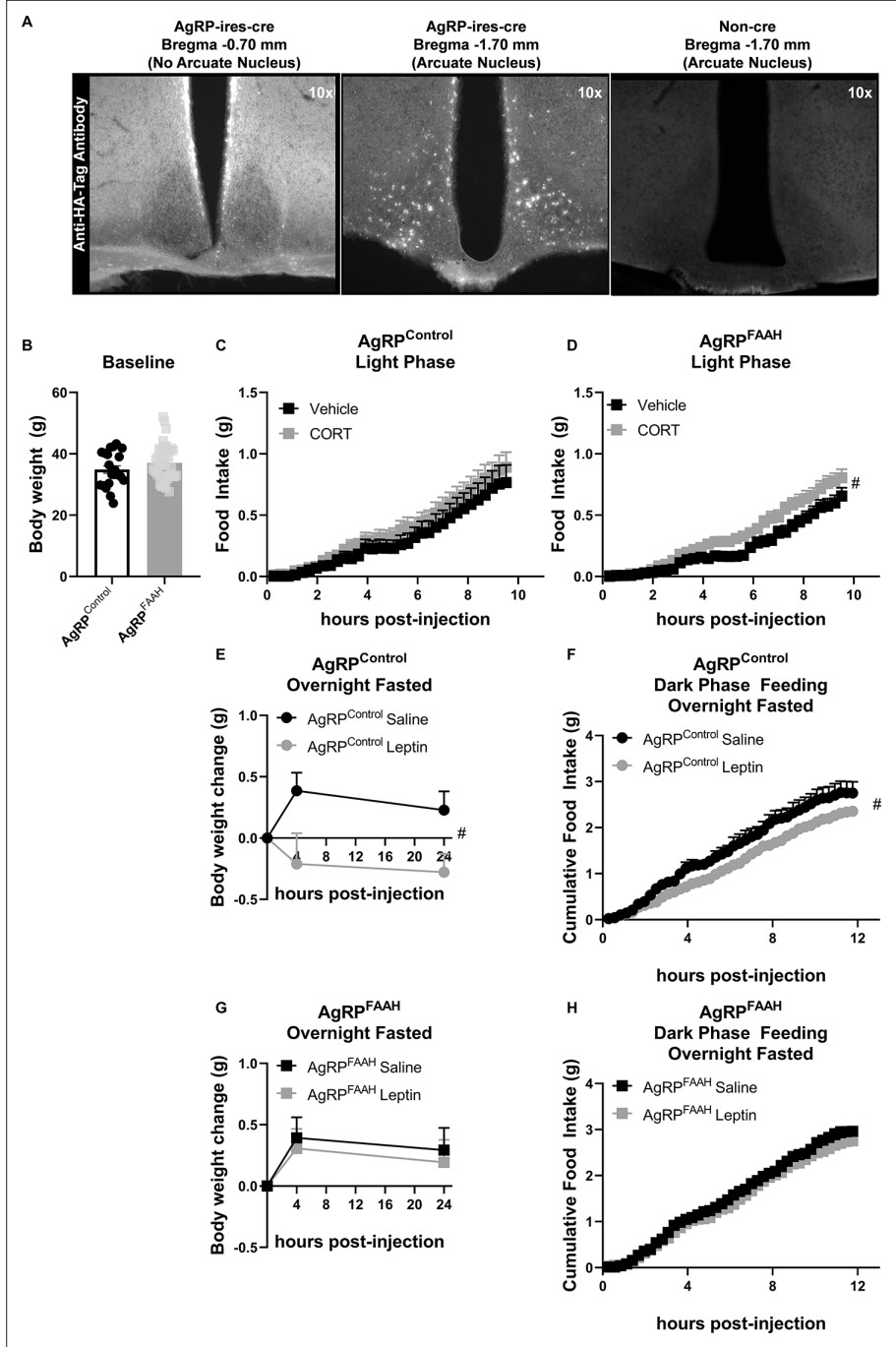

**Figure 5.** Fatty acid amide hydrolase (FAAH) knockdown in hypothalamic agouti-related protein (AgRP) neurons recapitulates the feeding responses to either acute corticosterone (CORT) or leptin treatment of FAAH A/A mice. (**A**) For validation of cre-dependent CRISPR/SaCas9 AAV-based knockdown of FAAH in AgRP neurons, HA-tag fluorescence was detected exclusively in the arcuate nucleus of AgRP-cre mice following bilateral injections of SaCas9-Control or SaCas9-FAAH into the hypothalamus of *AgRP-Ires-cre* mice. Non-cre mice showed no anti-HA-tag fluorescence in the arcuate nucleus. (**B**) Under basal conditions, there was no effect of FAAH knockdown in AgRP neurons on body weight (T(61) = 1.406, p=0.1648). (**C**) A single CORT injection (3 mg/kg) had no effect on cumulative light phase food intake in AgRP^control mice (treatment: F(1, 21.045)=0.465, p=0.503). (**D**) By contrast, acute CORT caused a hyperphagic feeding response in AgRP^FAAH mice measured during the light phase (treatment: F(1, 44.286)=4.257, p=0.045). (**E–F**) In terms of leptin exposure, AgRP^control mice were responsive to leptin-mediated reductions in weight gain (treatment: F(1, 22.253)=8.282, p=0.009) and refeeding (treatment: F(1,16.659)=5.635, p=0.030) following an overnight fast. (**G–H**) However, AgRP^FAAH mice were unresponsive to

*Figure 5 continued on next page*

*Figure 5 continued*

leptin (weight gain: F(1, 32.278=0.181, p=0.673); refeeding: F(1, 30.596)=0.703, p=0.408). For Panel (B), n=21 AgRP^control mice; n=42 AgRP^FAAH mice. For Panel (C), n=12 AgRP^control -vehicle; n=10 AgRP^control -CORT. For Panel (D), n=20 AgRP^FAAH-vehicle; n=21 AgRP^FAAH-CORT. For Panels (E and F), n=7 AgRP^control-saline; n=AgRP^control-leptin. For Panels (G and H), n=14 AgRP^FAAH-saline; n=15 AgRP^FAAH-leptin. Body weight and food intake data are presented as means ± SEM. Panel (B) was analyzed by Student's t-test. Panels (C–H) were analyzed by linear mixed models with repeated measures. Pound signs denote significant treatment effect. #p<0.05.

The online version of this article includes the following source data and figure supplement(s) for figure 5:

**Source data 1.** FAAH activity following mutagenesis of FAAH using SaCas9-FAAH in *CaMKIIa-cre* mice.

**Figure supplement 1.** FAAH activity following mutagenesis of FAAH using SaCas9-FAAH in CaMKIIa-cre mice.

GC and ghrelin orexigenic effects, which may suggest a generalized sensitivity of the FAAH A/A genotype to orexigenic signals, coupled to an impairment in the sensitivity to anorectic signals, such as leptin.

## Discussion

The ECS has an important role in the regulation of body weight. In terms of anandamide, activation of anandamide signaling is known to promote a positive energy balance and weight gain (*Mazier et al., 2015*). Moreover, endocannabinoid signaling lies downstream of several metabolic signals. For example, it is known that GCs and ghrelin, both orexigenic signals, mobilize endocannabinoid signaling, whereas anorexigenic leptin is known to suppress endocannabinoid signaling (*Balsevich et al., 2018*; *Bowles et al., 2015*; *Di Marzo et al., 2001*; *Kola et al., 2008*; *Tucci et al., 2004*). Therefore, genetic factors that modify endocannabinoid signaling may affect individual responses to orexigenic and anorexigenic signals. FAAH, the primary catabolic enzyme for anandamide, regulates anandamide signaling by controlling its levels. In humans, the common missense mutation (C385A) in FAAH affects FAAH protein stability, with the A/A genotype associated with approximately half the FAAH protein expression and enzymatic activity compared with the C/C genotype (*Chiang et al., 2004*). More specifically, T-lymphocytes from patients homozygous for the FAAH A/A variant express less than half of the FAAH protein and activity compared to WT lymphocytes. Here, we fully characterized the effects of the functional FAAH C385A mutation on GC-dependent metabolic outcomes in a novel C385A knock-in mouse model. Importantly, this FAAH C385A knock-in mouse model recapitulates the reduction in FAAH expression and activity observed in humans expressing the FAAH A/A variant (*Dincheva et al., 2015*). Using this model, we established the underlying molecular and cellular mechanisms that couple FAAH-AEA signaling to GC-induced feeding responses. We furthermore demonstrated that the FAAH C385A variant is not selective for GC-mediated metabolic responses, but rather governs an individual's sensitivity to additional orexigenic signals, namely ghrelin.

As a first step, we monitored the metabolic phenotype of FAAH C/C and A/A mice under basal conditions. In this condition, FAAH C/C and A/A mice did not present differences in body weight or body composition. Interestingly, the underlying metabolic parameters contributing to body weight regulation differed between FAAH C/C and A/A mice at baseline. Compared to FAAH C/C mice, FAAH A/A mice showed lower basal food intake and matching lower energy expenditures. This phenotype could be driven by basal elevations in AEA causing reduced basal metabolism, which in turn results in lower food intake to balance the alterations in energy expenditure. Alternately, in addition to AEA, FAAH also regulates the metabolism of fatty acid amides such as oleoylethanolamide, which is known to be a potent anorectic signal (*Gaetani et al., 2003*). Regardless of the exact mechanism of this subtle basal phenotype, this balance between food intake and energy expenditure offsets any differences in body weight or body composition at baseline. However, it may also predispose subjects with the FAAH A/A genotype to weight gain from overconsumption. Indeed, a low basal metabolic rate is a risk factor for weight gain in both humans and rodents (*Astrup et al., 1996*; *Astrup et al., 1999*; *Buscemi et al., 2005*; *Kunz et al., 2000*; *Lazzer et al., 2010*; *Sadowska et al., 2017*). Lower basal energy expenditure relates to lower basal maintenance costs, and consequently an energy imbalance could easily arise from overconsumption in response to specific environmental contexts (abundant access to palatable food) or endocrine states (elevated levels of orexigenic signals). In fact, this is in

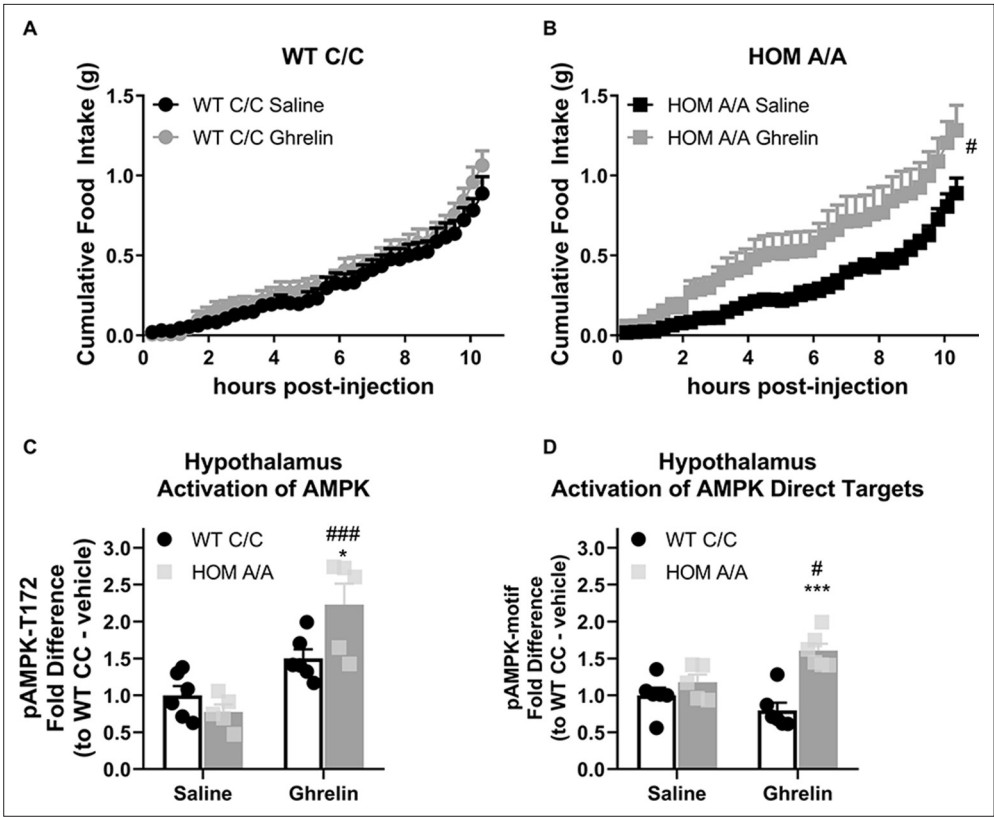

**Figure 6.** Fatty acid amide hydrolase (FAAH) A/A mice are hyperresponsive to the orexigenic effects of ghrelin. (**A–B**) Acute ghrelin (1 mg/kg, single i.p. injection) had no effect on cumulative food intake in FAAH C/C mice (treatment: F(1, 19.753)=0.494, p=0.490) yet increased light phase food intake in FAAH A/A mice (within the initial 10 hr following ghrelin injections) (treatment: F(1, 21.835)=5.406, p=0.030). (**C**) Hypothalamic AMP-activated protein kinase (AMPK) activity (as assessed by pAMPK protein expression) was significantly increased by ghrelin treatment exclusively in FAAH A/A mice (treatment × genotype: F(1, 18)=8.071, p=0.0108 plus post hoc testing). (**D**) Similarly, downstream targets of AMPK (with a shared consensus sequence) were activated from ghrelin exposure exclusively in FAAH A/A mice measured 1 hr following injections (treatment × genotype: F(1, 19)=9.597, p=0.0059 plus post hoc testing). For Panel (A), n=8 WT-saline; n=10 WT-ghrelin. For Panel (B), n=11 HOM-saline; n=10 HOM-CORT. For Panel (C), n=6 WT-saline biological replicates; n=6 WT-ghrelin biological replicates; n=5 HOM-saline biological replicates; n=5 HOM = ghrelin biological replicates. For Panel (D), n=6 WT-saline biological replicates; n=6 WT-ghrelin biological replicates; n=5 HOM-saline biological replicates; n=6 HOM-ghrelin biological replicates. Food intake data are presented as means ± SEM. Protein expression data are expressed as relative fold change compared to FAAH C/C-vehicle condition ± SEM. Panels (A–B) were analyzed by linear mixed models with repeated measures. Panels (C–D) were analyzed by two-way ANOVAs. Pound signs denote significant treatment effect. Asterisks denote significant genotype effect. *p<0.05, **p<0.01, #p<0.05, ###p<0.001.

The online version of this article includes the following source data and figure supplement(s) for figure 6:

**Source data 1.** The lastong effects of acute ghrelin treatment in FAAH C/C and A/A mice on metabolic outcomes.

**Figure supplement 1.** The lastong effects of acute ghrelin treatment in FAAH C/C and A/A mice on metabolic outcomes.

line with our findings that FAAH A/A mice show an amplified feeding response to GC and ghrelin orexigenic signals.

In terms of GC-endocannabinoid signaling interactions, it is recognized that the duration of GC exposure affects the interaction between them (reviewed by *Balsevich et al., 2017*). For example, GCs mobilize endocannabinoids through both rapid, non-genomic mechanisms and delayed, genomic mechanisms. Therefore, we examined three treatment durations to assess whether the FAAH C385A polymorphism is relevant to the chronic (4 weeks), immediate (48 hr), and/or acute (single exposure) metabolic outcomes of GCs. For chronic and immediate GC treatments, we delivered a low dose of CORT (25 µg/ml) in the drinking water, which maintains the diurnal rhythmicity of circulating GCs

(*Karatsoreos et al., 2010*). We found that regardless of treatment duration, GC exposure increased food intake exclusively in FAAH A/A mice. Furthermore, continuous GC exposure delivered for 48 hr, increased weight gain within the first 24 hr and 4 weeks of continuous GC exposure exaggerated GC-dependent weight gain. By contrast, there was no body weight phenotype 24 hr following acute GC injections, suggesting that continuous GC treatment is required to elicit weight gain. This agrees with the body of literature demonstrating that in the short term, GC actions are adaptive, redirecting energy stores and increasing food intake without affecting body weight maintenance (*Dallman et al., 2004*). However, prolonged GC exposure becomes maladaptive, as seen with chronic stress, which is associated with hypercortisolism, fat accumulation, and central obesity. In this context, our findings indicate that individuals with the FAAH A/A genotype are more susceptible to weight gain when confronted with elevations in GC signaling, and therefore may be particularly vulnerable to the influence of stress (*Pecoraro et al., 2004*; *Ulrich-Lai, 2016*).

The effects of the FAAH C385A variant on GC-mediated metabolic outcomes were specific to feeding responses. In particular, whereas the FAAH A/A genotype amplified the hyperphagic effects of GCs, there was no genotype effect on GC-dependent changes to energy expenditure or substrate utilization. Specifically, GC exposure decreased energy expenditure and increased carbohydrate utilization in both FAAH C/C and A/A mice. This agrees with the known role of GCs to suppress energy expenditure and promote carbohydrate utilization (*Dulloo et al., 1990*; *Flatt, 1993*; *Poggioli et al., 2013*). Therefore, our data suggest that while GCs mobilize AEA signaling to mediate feeding responses, it is unclear whether AEA signaling lies downstream of GC actions to regulate energy expenditure or substrate utilization. Regardless, the data herein delineate how the FAAH C385A variant affects energy balance by demonstrating that the FAAH C385A polymorphism modifies GC-induced weight gain through its effect on feeding rather than energy expenditure.

The exaggerated hyperphagic response to GC exposure was associated with increased hypothalamic anandamide levels exclusively in FAAH A/A mice. The hypothalamus is certainly recognized as one of the major brain regions responsible for endocannabinoid-mediated effects on feeding. Therefore, it is likely that elevated hypothalamic anandamide in the FAAH A/A genotype drives the heightened feeding response to orexigenic signals. Interestingly, one human study found that the C385A SNP did not associate with increased BMI per se, but rather higher levels of anandamide were positively associated with both the FAAH A/A genotype and a higher BMI (*Martins et al., 2015*). In agreement with this, we show that under basal conditions there is neither an effect of the A/A genotype on body weight nor anandamide content in brain regions central to endocannabinoid-mediated feeding. However, under the influence of specific endocrine drivers, such as GCs, anandamide levels increased exclusively in FAAH A/A mice within the hypothalamus, which likewise associated with a hyperphagic response selectively in the FAAH A/A mice. Taken together, GCs elevate hypothalamic anandamide levels selectively in the FAAH A/A genotype to sensitize the FAAH A/A genotype to hyperphagic feeding outcomes.

We next sought to characterize the cellular population within the hypothalamus driving the AEA-dependent exaggerated feeding response to GC exposure. Within the arcuate nucleus of the hypothalamus, POMC- and AgRP-expressing neurons are considered the prototypic regulators of homeostatic food intake (*Gautron et al., 2015*). Whereas POMC neuronal activity promotes satiety, AgRP neuronal activity promotes feeding. Interestingly, GCs increase AgRP neuronal firing rates to induce hyperphagia (*Perry et al., 2019*). Therefore, we examined whether FAAH knockdown exclusively in AgRP neurons is sufficient to recapitulate the exaggerated feeding response to GC exposure seen in FAAH A/A mice. For this purpose, we employed a recently described conditional single vector CRISPR/SaCas9 virus system (*Hunker et al., 2020*) for efficient cre-dependent mutagenesis of *Faah* (or control) into *AgRP-Ires-cre* mice. Considering there is low FAAH activity in the sparsely distributed hypothalamic AgRP neurons, it was unfeasible to functionally measure changes in AgRP FAAH activity, driven by the saCas9-FAAH construct, using an enzymatic FAAH activity assay. Therefore, to confirm cre-dependent reductions of FAAH activity on account of our SaCas9-FAAH construct, we assessed FAAH enzymatic activity in the hippocampus where FAAH expression is high (*Egertová et al., 1998*; *Gulyas et al., 2004*; *Tsou et al., 1998*). Following targeted intra-hippocampal delivery of the SaCas9-FAAH construct in *CaMKIIα-cre* mice to drive cre expression in hippocampal CA1 pyramidal cells (*Tsien et al., 1996*), we measured a dramatic, significant reduction in FAAH activity compared to the SaCas9-Control vector, validating the construct. Remarkably decreasing FAAH expression selectively

in AgRP neurons paralleled the feeding phenotype of FAAH A/A mice in response to acute CORT injections. Specifically, under vehicle conditions AgRP[FAAH] mice displayed reduced cumulative food intake despite no difference in basal body weights compared to AgRP[Control] mice. However, following a single injection of CORT, AgRP[FAAH] mice increased their light phase feeding, a period when feeding is typically low. By contrast, this hyperphagic response to acute CORT was absent in Agrp[Control] mice.

Since there are no reliable FAAH antibodies to detect FAAH expression nor is there a fluorophore in the *AgRP-Ires-cre* mice, we could not quantify the percent of AgRP neurons targeted by our viral delivery system. Therefore, it is possible that only a subset of AgRP neurons were targeted and lacked FAAH. In fact, our HA-tagged SaCas9 protein expression profile, used to confirm cre-mediated DNA recombination, supports the notion that a discrete subset of AgRP neurons was targeted. Specifically, the HA-tagged SaCas9 protein expression spread into the dorsal aspects of the arcuate nuclei whereas AgRP neurons are largely concentrated in the ventral aspects of the arcuate nuclei. This suggests that indeed only a subset of AgRP neurons were targeted. It is possible that we targeted more dorsally positioned AgRP neurons based on our injection strategy wherein we positioned the tip of the cannula toward the dorsal aspects of the arcuate nuclei to avoid penetrating the base of the brain and cerebrospinal fluid. Nevertheless, this study demonstrates that lowering FAAH expression exclusively in AgRP neurons (targeted or widespread) is sufficient to recapitulate the hyperphagic feeding response of FAAH A/A mice to acute GC treatment. Of course, both GCs and AEA act broadly across several cell types and brain regions to regulate feeding. Our findings that reducing FAAH expression in AgRP neurons heightens GC-mediated feeding does not preclude the involvement of other brain regions. Whether other neuronal populations are likewise involved will require further work.

Given the effects of FAAH knockdown in AgRP neurons on GC-dependent feeding responses, we investigated whether AgRP neurons are likewise important cellular mediators of FAAH's effects on leptin-dependent feeding outcomes. It is known that leptin suppresses hypothalamic endocannabinoid signaling to decrease feeding (*Balsevich et al., 2018*; *Cardinal et al., 2012*; *Di Marzo et al., 2001*; *Jo et al., 2005*). Furthermore, we have specifically shown that elevated AEA signaling in FAAH A/A mice prevents the anorectic effects of leptin following 16 hr food deprivation (*Balsevich et al., 2018*). Therefore, we hypothesized that FAAH deletion specifically in AgRP neurons would dampen the hypophagic responses to leptin, similar to what was seen in FAAH A/A mice. In agreement with this, AgRP[Control] mice exhibited weight loss and reduced feeding in response to acute leptin exposure. However, AgRP[FAAH] mice were unresponsive to acute leptin. Our results indicate that AgRP neurons are an important cellular population mediating the effects of FAAH-AEA signaling events of multiple upstream endocrine signals regulating feeding behavior.

We next explored whether the FAAH C385A variant modulates the metabolic outcomes of additional orexigenic signals. Specifically, we examined the metabolic phenotypes of FAAH C/C and A/A mice in response to ghrelin exposure based on the known interactions between ghrelin and endocannabinoid signaling pathways (*Edwards and Abizaid, 2016*). A single, subthreshold dose of ghrelin paralleled the outcomes arising from acute GC exposure, wherein ghrelin increased light phase food intake exclusively in FAAH A/A mice. Again, the effects were specific to feeding as ghrelin had no effect on energy expenditure across genotypes. Considering chronic stress elevates circulating GC and ghrelin levels (*Abizaid, 2019*), it is reasonable to suspect that individuals presenting the FAAH A/A genotype are more susceptible to stress-associated weight gain on account of amplified responses to these orexigenic signals. Follow-up studies should address the feeding response of FAAH C/C and A/A mice to acute and chronic stress exposures.

An important question raised by this study is how does the FAAH C385A variant coordinate the feeding responses to multiple orexigenic signals? For instance, is there a common pathway downstream of FAAH-AEA signaling driving the orexigenic feeding responses? Along these lines, it has previously been shown that GCs and ghrelin activate hypothalamic AMPK through a CB1R-dependent mechanism (*Kola et al., 2008*; *Scerif et al., 2013*). Indeed, hypothalamic AMPK activity elicits powerful feeding responses (*Minokoshi et al., 2004*). In agreement with this, we found that AMPK signaling is heightened in the hypothalamus following acute exposure to either GCs or ghrelin exclusively in FAAH A/A mice. Moreover, pretreatment with an AMPK inhibitor blocked the orexigenic response to acute GC treatment in FAAH A/A mice. The data herein provide unequivocal evidence that AMPK lies downstream of FAAH-AEA signaling to coordinate ghrelin and GC signaling.

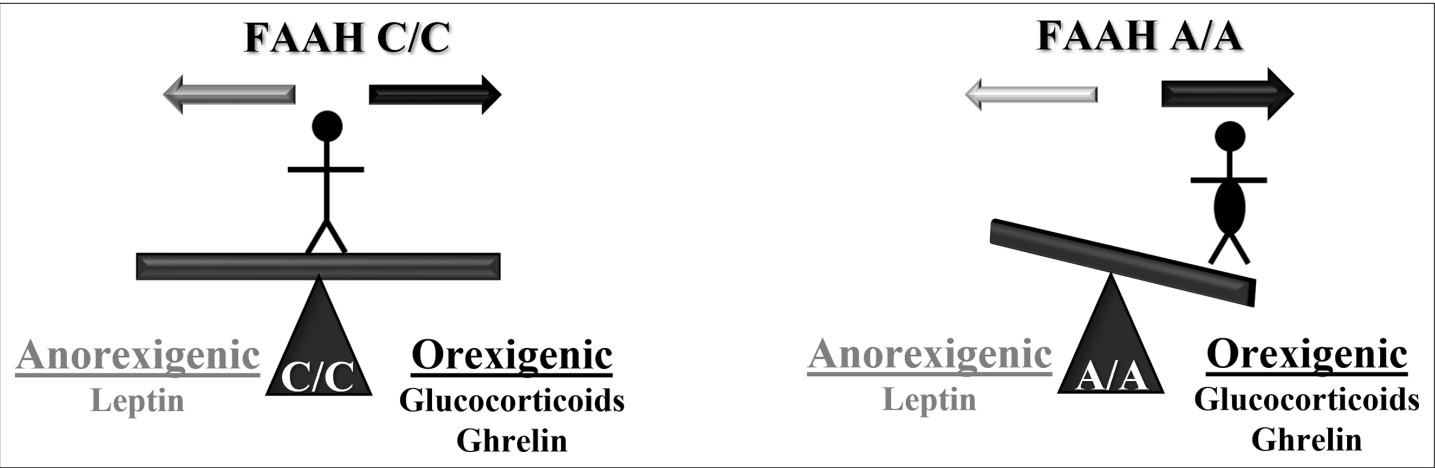

**Figure 7.** Working model of how the fatty acid amide hydrolase (FAAH) C385 polymorphism influences individual susceptibility to body weight gain. Individuals with the FAAH C/C genotype are able to respond to anorexigenic and orexigenic signals based on their metabolic needs. However, the FAAH A/A genotype shifts responses to feeding cues, sensitizing individuals to orexigenic signals, favoring weight gain, and desensitizing individuals to weight loss.

The current study clearly demonstrates that the FAAH A/A genotype predisposes individuals to the feeding responses of orexigenic signals, CORT and ghrelin. Specifically, our working model is that orexigenic signals mobilize AEA in AgRP neurons of the hypothalamus. This effect is amplified in the A/A genotype (with low FAAH expression) because there is reduced enzymatic degradation of AEA. The exaggerated AEA levels lead to heightened CB1R activation and downstream AMPK activation to ultimately sensitize A/A individuals to orexigenic signaling outcomes. By contrast, our previous work determined that FAAH A/A mice are less responsive to the anorexigenic signal leptin (*Balsevich et al., 2018*). Furthermore, the current study established that FAAH knockdown exclusively in AgRP neurons is sufficient to blunt feeding and body weight responses to leptin. Thus, based on the current study and our previous findings, we demonstrate that the FAAH C385A variant shifts an individual's sensitivity to metabolic feeding signals, where individuals with the FAAH A/A genotype are more responsive to orexigenic signals at the expense of being less responsive to anorexigenic signals (*Figure 7*). Given orexigenic signals favor weight gain and anorexigenic signals favor weight loss, individuals with the minor A/A genotype may show exaggerated weight gain in response to environmental conditions that increase orexigenic signals (such as GCs and ghrelin) and decrease anorexigenic signals (such as leptin). Chronic stress, for example, is known to increase levels of both GCs and ghrelin while also producing decreased levels of leptin (*Bouillon-Minois et al., 2021*; *Lu et al., 2006*). Collectively, our data indicate that human studies be conducted for the FAAH C385A polymorphism in a manner that considers external environmental contexts.

## Methods
### Animals
The FAAH C385A knock-in mouse line (MGI # 5644434) had been previously generated and characterized (*Dincheva et al., 2015*). The *Agrp-Ires-cre* mice (JAX #012899) and the CamKIIα-cre mice (JAX # 005359) were purchased from Jackson Laboratories (Bar Harbour, Maine). For all experiments, age-matched male mice between 3 and 6 months were used. All mice were group-housed until 1–2 weeks before the experiment onset at which point mice were individually housed. For the duration of the experiments, mice were maintained under controlled temperature (25±1°C) and humidity (30±5%) conditions. For acute studies with CORT and ghrelin, mice were maintained on a 12:12 hr light/dark cycle with lights on at 08:00. For extended exposure studies, mice were maintained on a 12 hr reverse dark/light schedule, with lights on at 22:00. For acute leptin study, mice were also maintained on a 12 hr reverse dark/light schedule with lights on at 22:00. Mice received ad libitum access to tap water and standard lab chow. Although sample size calculations were not performed, sample sizes for each animal study were determined from prior studies in our laboratory examining metabolic outcomes

(feeding and energy expenditure) in mice (*Balsevich et al., 2018*; *Bowles et al., 2015*). Mice of each genotype were randomly assigned to a treatment group counterbalanced for body weight. Blinding was not employed. The experimental unit for all studies was a single adult, male animal. All studies were carried out in compliance with the ARRIVE guidelines. All protocols had been approved by the University of Calgary Animal Care Committee and were carried out in accordance with Canadian Council on Animal Care.

### Chemical reagents

CORT (Sigma, 27840), human ghrelin (Tocris, 1463), recombinant mouse leptin (Cedarlane Laboratories Ltd, CLCYT351-2), and compound C (Selleck Chemicals, S7306) were used in all in vitro and in vivo experiments. All reagents were prepared freshly immediately before the onset of use.

### Extended CORT exposure

For 48 hr and 30-day CORT exposure, mice were given CORT (25 µg/ml) or vehicle (1% ethanol) through their drinking water as previously described (*Karatsoreos et al., 2010*). Briefly, treatment began at the onset of their dark (active phase) and continued for either 48 hr or 30 days. Metabolic measurements were monitored throughout the treatment period. For 30-day treatment, fresh CORT/ vehicle-treated water was supplied every 3–4 days. At the end of the experimental period, mice were sacrificed by rapid decapitation at the onset of the dark phase. Tissues were rapidly frozen and stored at –80°C until further processing.

### Acute exposure studies

For acute exposure, mice were administered with CORT (3 mg/kg) or vehicle (saline/DMSO/Tween-80, 18:1:1) by intraperitoneal injection (i.p.). Similarly, ghrelin (1 mg/kg) or vehicle (saline) and leptin (2 mg/kg) or vehicle (saline) were administered by i.p. injection. For CORT and ghrelin studies, drug administration occurred 2 hr after the onset of the light (inactive) phase to capture hyperphagic responses in light phase food consumption. For leptin study, drug administration occurred at the onset of the dark phase following an overnight fast (~16 hr). After 24 hr of metabolic measurements, mice were allowed to recover in their home cages for at least 1 week before sacrifice. On the day of sacrifice, mice were administered with their respective drug 1 hr prior to rapid decapitation and tissue collection. Harvested tissues were stored at –80°C until further processing.

### Pharmacological blockade of AMPK

Compound C (5 mg/kg), an AMPK inhibitor, or vehicle (saline) were administered by i.p. injections 1 hr before injecting CORT (3 mg/kg) or vehicle to mice according to the 'Acute exposure studies' section above. Briefly, mice received compound C or saline at the onset of the light phase. One hr later, mice received a second injection of either CORT or vehicle in a 2×2 design. Metabolic measurements were monitored over the next 24 hr.

### Generation of AAV1-CMV-FLEX-HA-Tag, SaCas9-U6-sgRNA

The single guide (sg)RNA for targeting exon 1 of *Faah* was designed as previously described (*Hunker et al., 2020*). Briefly, the sgRNA sequence targeting exon 1 of *Faah* (5'- CTGCAGGCTAGGCAAA CC-3') and a control sgRNA sequence (5'- CTGCAGGCTAGGCAAACCTTT-3') were synthesized (Invitrogen) and cloned into the shuttle plasmid for adeno-associated viral (pAAV-FLEX-SaCas9-U6-sgRNA; Addgene #124844) to generate a plasmid for cre-dependent expression of HA-tagged *Staphylococcus aureus* Cas9 (SaCas9) and selective mutagenesis of *Faah* (SaCas9-FAAH) (*Hunker et al., 2020*). The control AAV vector (SaCas9-Control) had an identical guide except the three terminal base pairs in the seed region of the guide sequence were mutated. Following sequence validation, AAV serotype 1 (AAV1) vectors were generated for SaCas9-FAAH and SaCas9-Control using the packaging plasmid pDG1 and transient transfection of HEK293T cells, as described (*Hunker et al., 2020*).

### Stereotaxic surgical procedure

Mice were anesthetized with isofluorane and placed on a stereotaxic frame. For FAAH knockdown in AgRP neurons, a cre-dependent CRISPR/Cas9 AAV vector targeting *Faah* (SaCas9-FAAH) or control (SaCas9-Control) was delivered bilaterally (210 nl per side) into the arcuate nucleus of the hypothalamus

(AP: –1.6 mm, ML: ±0.15 mm, DV: –5.8 mm). For FAAH knockdown in CamKIIα-cre mice, SaCas9-FAAH or SaCas9-Control was bilaterally injected (400 nl per side) into the dorsal hippocampus (AP: –2.1 mm, ML: ±1.3 mm, DV: –1.4 mm). Virus was injected at a rate of 0.1 µl min$^{-1}$. Metacam (3 mg/kg) was given as a postoperative analgesic. Animals recovered for a minimum of 4 weeks in their home cage to allow for viral expression before the onset of experimentation.

## FAAH activity

To assess FAAH activity, hippocampi were collected from CamKIIα mice that either received SaCas9-FAAH or SaCas9-Control vector. Briefly, hippocampi were excised on ice and immediately snap frozen and stored at –80°C. Hippocampi were homogenized and membrane fractions were isolated as previously described (*Gray et al., 2015*). Enzymatic activity of FAAH was measured as the conversion of [$^3$H]-labeled AEA (in the ethanolamine portion of the molecule) to [$^3$H]ethanolamine as previously described (*Maccarrone et al., 1999*). Membranes were incubated in TME buffer (50 mM Tris-HCl, 3.0 mM MgCl$_2$, and 1.0 mM EDTA, pH 7.4) containing 1.0 mg/ml fatty acid-free bovine serum albumin and 0.2 nM [$^3$H]AEA. Calibration curves were prepared using eight concentrations of AEA at concentrations between 10 nM and 10 µM. Incubations were kept at 37°C and were stopped by adding ice-cold chloroform/methanol (1:2). Samples were incubated at room temperature for 30 min before the addition of chloroform and water. Aqueous and organic phases were separated by centrifugation at 1000 rpm for 10 min. The concentration of [$^3$H] in the aqueous phase was determined by liquid scintillation counting and the conversion of [$^3$H]AEA to [$^3$H]ethanolamine was calculated. The maximal hydrolytic activity (Vmax) for this conversion was determined by fitting the data to the Michaelis-Menten equation in Prism v8 (GraphPad, San Diego, CA, USA).

## Metabolic measurements

Energy expenditure (heat), food intake, and RER (=CO$_2$ produced/O$_2$ consumed) were measured by indirect calorimetry in Comprehensive Lab Animal Monitoring System (CLAMS) metabolic chambers (Columbus Instruments; Columbus, OH, USA) as previously described (*Pezeshki et al., 2015*). Following 2–4 days of acclimatization to the chambers, metabolic measurements were performed every 14 min over the experimental period (flow rate 0.5 L min$^{-1}$). TEE was calculated using O$_2$ consumption (VO$_2$, [ml h$^{-1}$]) and CO$_2$ production (VCO$_2$, [ml h$^{-1}$]) in the following equation following a previously published protocol (*Pezeshki et al., 2015*): Calorific value = VO$_2$ × [3.815 + (1.232×RQ)], with data collection and processing using Oxymax v5.2 and CLAX v2.2.2. Data are represented as kcal h$^{-1}$. When mice spilled their food, their food intake was excluded from analyses (see Supplementary source data). Body composition was examined by quantitative magnetic resonance using the Minispec LF-110 NMR Analyzer (Bruker Optics, Milton, ON, Canada) on awake mice.

## Home-cage activity

Mice were surgically implanted with a telemetric transponder (Data Sciences International, DSI, St. Paul, MN, USA) to examine activity counts in their home-cages. Briefly, under isoflurane anesthesia, a small incision was made along the midline of the abdomen. The abdominal wall was opened and the transponder was implanted into the abdominal cavity of the mice. The incision was closed with absorbable sutures and the mice were allowed to recover for 1 week before home-cage activity counts were measured under basal conditions and during 24 hr CORT exposure (delivered in their drinking water). Metacam (3 mg/kg) was given as a postoperative analgesic.

## Endocannabinoid quantification

Lipids were extracted from harvested tissues as described previously (*Hill et al., 2009*). Briefly, weighed tissue samples were homogenized in borosilicate glass culture tubes containing 2 ml of acetonitrile with 84 pmol of [$^2$H8]anandamide. Tissues were subsequently sonicated for 30 min. To precipitate proteins, samples were incubated overnight at –20°C proteins, followed by centrifugation at 1500 × $g$ to remove particulates. The supernatants were collected and evaporated to dryness using N$_2$ gas. Next, samples were suspended in methanol and dried using N$_2$ gas. The resulting lipid extracts were suspended in methanol and stored at –80°C until further processing. AEA tissue levels were subsequently determined using isotope-dilution, liquid chromatography-mass spectrometry as described previously (*Patel et al., 2005*).

## CORT quantification

Plasma CORT levels were determined using a commercially available enzyme immunoassay kit (Arbor Assays, Ann Arbor, MI, USA, Catalogue Number K014-H5; sensitivity 20.9 pg/ml) according to the manufacturer's instructions.

## Cell lines and culture conditions

The GT1-7 hypothalamic cell line (RRID:CVCL_0281) was kindly provided by Dr. Pamela Mellon (Salk Institute, La Jolla, CA, USA). Cells were grown in Dulbecco's modified Eagle's medium (DMEM) supplemented with 10% fetal bovine serum, 25 mM glucose, and 1% penicillin/streptomycin at 37°C in 95% $O_2$ and 5% $CO_2$. Once cells reached ~80% confluence, experiments were initiated. Briefly, cells were starved for 4 hr in DMEM supplemented with 1 mM glucose. During this incubation period, FAAH inhibitor URB597 (1 µM for 4 hr) or vehicle (DMSO for 4 hr) was added to the media. To examine the effects of CORT and CORT×FAAH interactions on AMPK activity, CORT (1 µM) or vehicle (DMSO) was added to the media for an additional 2 hr. After 6 hr, cells were immediately trypsinized, centrifuged, and stored as pellets at –80°C until further processing.

## Capillary-based immunoblotting of proteins

Tissues or cells were homogenized in Pierce IP Lysis buffer supplemented with protease inhibitor cocktail 3 (Merck Millipore, Darmstadt, Germany) and phosphatase inhibitor PhosSTOP (Roche, Penzberg, Germany). Following centrifugation, the cleared supernatant (total protein extract) was collected. Samples were subsequently prepared for the automated capillary-based immunoblotting system, WES (ProteinSimple, Bio-Techne, San Jose, CA, USA) according to the manufacturer's instructions. Target proteins were probed with the following primary antibodies: AMPKα (1:100, Cell Signaling Technology, #2532, RRID:AB_330331), pAMPK (T172) (1:50, Cell Signaling Technology, #2531, RRID:AB_330330), pAMPK substrate motif (1:50, Cell Signaling Technology, #5759, RRID:AB_10949320), and secondary antibody anti-rabbit IgG, HRP-linked antibody (1:10,000, Cell Signaling Technology, #7074, RRID:AB_2099233). Total protein was quantified using the protein normalization module (Protein-Simple, Bio-Techne, San Jose, CA, USA, #DM-PN02). Signal intensities of phosphorylated proteins were normalized to their respective non-phosphorylated proteins. According to standard practice, virtual bands are not presented since obtained results are signal intensities in a digital format. Quantitative analysis was performed using the Compass software (Protein Simple, Bio-Techne, San Jose, CA, USA) and data expressed as fold change relative to control group.

## Immunohistochemistry

For validation of cre-dependent CRISPR/SaCas9 AAV-vector expression, HA-tag fluorescence HA-tagged SaCas9 was visualized using the following antibodies: rabbit anti-HA-tag (Sigma-Aldrich, #H6908, RRID:AB_260070) and secondary antibody anti-rabbit Alexa Fluor 488 (1:1000, Thermo Fisher, # 32731, AB_2633280). Briefly, mice were deeply anesthetized with pentobarbitol before undergoing intracardial perfusion. Mice were perfused with PBS before being fixed with a 4% paraformaldehyde solution (PFA). Brains were then removed and dropped into a 4% PFA solution for 24 hr. Following drop fixing, brains were placed into a 30% sucrose solution for 48 hr for cryoprotection. Fixed brains were frozen and sliced at 40 µm thickness on a microtome and stored in an anti-freeze solution until they were ready to be stained. Before staining for HA, slices were washed three times in PBS and three times in PBST. Following washing steps, slices were stained in a 1:1500 dilution of anti-HA for 24 hr at 4°C. Following staining, slices were washed in PBST three more times, then stained with a 1:200 dilution of Alexa Fluor 488 for 1 hr. Slices then underwent three more washes in PBST before being mounted on glass slides and covered with Fluoroshield with DAPI stain (Sigma F6057). Mounted slices were imaged using a fluorescent microscope (Leica DM4000 B LED), and CRISPR expression was imaged under an L5 filter cube. Brightness and contrast were adjusted equally across all images in ImageJ.

## Statistical analysis

Data were analyzed using IBM SPSS Statistics 26 software (IBM SPSS Statistics, IBM, Chicago, IL, USA). Independent t-tests were used to compare genotype effects. Two-way ANOVAs (analysis of variance) were used to examine studies with a 2×2 design. Where the initial test yielded a significant interaction,

independent Student's t-tests (two-tailed) were conducted to locate the interaction effect using simple comparisons. Repeated measures on CLAMS data (energy expenditure, respiratory quotient, and food intake) were analyzed by linear mixed models. The fixed effects of treatment, genotype, time, and their interactions were included in the model. Animal nested in the group was the random variable on which repeated measures were taken and covariance structures modeled (*Pezeshki et al., 2015*). Body weight was included as a covariate in the analyses of energy expenditure (*Tschöp et al., 2011*). Post hoc analyses were undertaken using Bonferroni's multiple comparison post hoc test where appropriate. Statistical analyses of CLAMS data were performed using individual data points (every 15 min) collected over the experimental period. For 24 hr measurements, CLAMS data were graphed using the four-point averages of individual data points. Outlier testing was performed using Grubb's test for outliers, and when an outlier was detected ($p < 0.05$), it was omitted from analysis. For the 48 hr CORT exposure study, select metabolic chambers (see Supplementary source data) had dysfunctional $O_2/CO_2$ sensors and were excluded from RER and TEE analyses. Statistical significance was set at $p < 0.05$; a statistical tendency was set at $p < 0.1$. For interactions at $p < 0.1$, we also examined lower order main effects. Data are presented as the mean ± SEM.

## Acknowledgements

The authors thank Dr. Frank Visser for his genotyping support. We acknowledge Dr. Pamela Mellon for donating the GT1-7 cells. We thank the animal care staff at the University of Calgary for providing technical support. This work was supported by funding from the Natural Sciences and Engineering Research Council of Canada to MNH; American Heart Association Grant # 953881 to PKC. GB received salary support from a Banting Postdoctoral Fellowship provided by the Canadian Institutes of Health Research (CIHR) and an Alberta Innovates Scholarship.

## Additional information

### Funding

| Funder | Grant reference number | Author |
|---|---|---|
| Natural Sciences and Engineering Research Council of Canada | | Matthew N Hill |
| American Heart Association | 953881 | Prasanth K Chelikani |
| Canadian Institutes of Health Research | | Georgia Balsevich |
| Alberta Innovates | | Georgia Balsevich |

The funders had no role in study design, data collection and interpretation, or the decision to submit the work for publication.

### Author contributions

Georgia Balsevich, Conceptualization, Data curation, Formal analysis, Investigation, Methodology, Writing – original draft, Writing – review and editing; Gavin N Petrie, Daniel E Heinz, Investigation, Methodology; Arashdeep Singh, Investigation, Writing – review and editing; Robert J Aukema, Hiulan Yau, Martin Sticht, Investigation; Avery C Hunker, Methodology; Haley A Vecchiarelli, Validation, Investigation; Roger J Thompson, Francis S Lee, Resources; Larry S Zweifel, Prasanth K Chelikani, Resources, Methodology, Writing – review and editing; Nils C Gassen, Investigation, Methodology, Writing – review and editing; Matthew N Hill, Conceptualization, Supervision, Funding acquisition, Methodology, Project administration, Writing – review and editing

### Author ORCIDs

Georgia Balsevich https://orcid.org/0000-0002-7839-018X
Daniel E Heinz https://orcid.org/0000-0001-7103-9621
Arashdeep Singh https://orcid.org/0000-0002-2477-1438

Robert J Aukema http://orcid.org/0000-0003-3490-3390
Haley A Vecchiarelli http://orcid.org/0000-0001-6331-6107
Roger J Thompson http://orcid.org/0000-0002-7019-7246
Francis S Lee http://orcid.org/0000-0002-7108-9650
Larry S Zweifel http://orcid.org/0000-0003-3465-5331
Matthew N Hill http://orcid.org/0000-0001-7144-9209

### Ethics

All studies were carried out in compliance with the ARRIVE guidelines. All protocols had been approved by the University of Calgary Animal Care Committee and were carried out in accordance with Canadian Council on Animal Care (under protocols AC16-0171, AC16-0053, AC20-0003, and AC20-0090). All surgery was performed under Isofluorane anesthesia and Metacam was given as a post-operative analgesic.

### Decision letter and Author response

Decision letter https://doi.org/10.7554/eLife.81919.sa1
Author response https://doi.org/10.7554/eLife.81919.sa2

## Additional files

### Supplementary files

• MDAR checklist

### Data availability

All data generated or analysed during this study are included in the manuscript and supporting files. Source data files have been provided for all figures and supplemental figures.

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
