## [Editor Report]

This study work provides evidence that an enzyme in key neurons in the brain regulate body weight. They used novel mouse models to mimic mutations in this gene in humans. The work is significant as it reconciles previously contradictory clinical data. Thus, the studies will be of wide interest.

---

## [Decision Letter]

**Decision letter after peer review:**

Thank you for submitting your article "A Genetic Variant of Fatty Acid Amide Hydrolase (FAAH) Exacerbates Hormone- Mediated Orexigenic Feeding in Mice" for consideration by *eLife*. Your article has been reviewed by 2 peer reviewers, and the evaluation has been overseen by a Reviewing Editor and Michael Taffe as the Senior Editor. The following individuals involved in review of your submission have agreed to reveal their identity: Sachin Patel (Reviewer #1); Streamson Chua (Reviewer #2).

Essential revisions:

1) Body weight data over the lifespan of the FAAH variants is not shown, an important basis for the work is that basal differences do not exist in the FAAH mutation mice, but data are only shown for one point in time. Without some longitudinal assessment of body weight this conclusion is not supported.

2) I do not see full blots for Figure 4. Bands should be shown in the figures with full blots in the supplement. It would also be important to note the methodology in the Results section, I had to dig into the methods to figure out this was a WB assay rather than an ELISA or other method.

3) In figure 2 the N or 4-5 is not convincing that this is a robust and reproducible effect. This is also highlighted by the trend level significance via ANOVAs in figure 2.

4) Hypothalamic measures of AEA and AMPK activity would suggest that AGRP neurons might play a major role in determining the measurements. However, the viral injections would suggest that only a minor fraction of AGRP neurons were targeted by the viral system. Typically, the greatest concentration of AGRP neurons are in the ventromedial aspect of the ARC whereas the HA tag is sparsely distributed over the more lateral and dorsal aspect of the ARC. An examination of the DROP SEQ data for the ARC by the Lowell/Tsai lab would indicate that there is some variability in the expression of FAAH in AGRP neurons.

5) Some mention of the similarities of the mouse and human Faah alleles would be appropriate – in terms of effects on the FAAH steady state concentrations and impact on activity.

*Reviewer #1 (Recommendations for the authors):*

"The ECS, and specifically anandamide signaling, has an important role in the regulation of body weight."- there is just as much evidence to support 2-AG signaling in the regulation of body weight… maybe more. This should be adjusted.

*Reviewer #2 (Recommendations for the authors):*

The HA-tag imaging of Cas9 expression does not recapitulate AGRP neuron distribution within the ARC, which is concentrated in the ventromedial aspects of the two ARC nuclei whereas the HA-tag labeling shows a very diffuse and sparse distribution within the dorsolateral aspects of the ARC as well as the spaces between the ARC and the VMN.

---

## [Author Response]

Essential revisions:1) Body weight data over the lifespan of the FAAH variants is not shown, an important basis for the work is that basal differences do not exist in the FAAH mutation mice, but data are only shown for one point in time. Without some longitudinal assessment of body weight this conclusion is not supported.

We thank the reviewer for pointing out this omission. We agree that it is important to examine body weight at different time points across the lifespan. Our animals are bred in a separate facility and we do not gain access to the animals until ~2-4 weeks before the onset of the experiments. Nevertheless, we compiled our body weight data across all experiments to examine body weights at 2-, 3-, and 4- months of age under basal (home cage) conditions. Regardless of their age, FAAH C/C and FAAH A/A mice show no difference in body weight under basal conditions. We have added these results to the revised manuscript as Supplemental Figure S1.

2) I do not see full blots for Figure 4. Bands should be shown in the figures with full blots in the supplement. It would also be important to note the methodology in the Results section, I had to dig into the methods to figure out this was a WB assay rather than an ELISA or other method.

For protein analysis we used capillary-based immunoblotting (BioTechne, ProteinSimple, WES) a novel technique that allows automated separation and detection of proteins from lysates with very low amounts. This system is based on automated quantification of luminescence signals generated from horseradish-peroxidase secondary antibodies, similar to classical western blots, but more sensitive. Obtained results are signal intensities in a digital format. Usually, virtual bands are not presented for that method. We added more info in the methods section for clarification and titled this section “Immunoblotting of Proteins” to avoid confusion.

3) In figure 2 the N or 4-5 is not convincing that this is a robust and reproducible effect. This is also highlighted by the trend level significance via ANOVAs in figure 2.

For the CLAMS data (Figures 2D – 2I), the sample size of n = 4-5 is sufficient to achieve robust effects based on the repeated measurements taken ~ every 15 minutes on an individual mouse over almost 24 h (~80 measurements per mouse). The statistics affirm the robustness. With regards to Figure 2A, the reviewer pointed out that we detected a trend for body weight change following 4 weeks of CORT treatment. Specifically, we ran a 2-way ANOVA with genotype and treatment as independent factors. We detected a main effect of CORT treatment as well as a genotype x treatment trend (F(1,15) = 4.456, p = 0.052). As indicated in the ‘Statistical Analysis’ subsection embedded in the Methods, we examined lower order main effects for interactions at p < 0.1 in order to understand what is driving the interaction. Using this methodology, we found that CORT-treated HOM A/A mice gained significantly more weight than vehicle-treated HOM A/A mice, whereas CORT-treated WT C/C mice did not gain significantly more weight than vehicle-treated WT C/C mice. This agrees with our CLAMS data that CORT treatment significantly increased feeding in HOM A/A mice but not in WT C/C mice. As a follow-up to the reviewer’s concern about Figure 2A’s interpretation, we performed a cohen’s d-test to assess the effect size for the t=T-test between vehicle-treated and CORT-treated HOM A/A mice.

The cohen’s d-test:

Cohen's d = (M2 – M1) ⁄ SD_pooled_

Cohen's *d* = (4.12 – 0.7) ⁄ 0.869511 = 3.933244.

Based on the cohen’s d-test, the effect size is large. Together, based on the effect size and agreement between food intake and changes in body weight, we are confident in the robustness of our data and our interpretation of the genotype x treatment trend for body weight gain.

4) Hypothalamic measures of AEA and AMPK activity would suggest that AGRP neurons might play a major role in determining the measurements. However, the viral injections would suggest that only a minor fraction of AGRP neurons were targeted by the viral system. Typically, the greatest concentration of AGRP neurons are in the ventromedial aspect of the ARC whereas the HA tag is sparsely distributed over the more lateral and dorsal aspect of the ARC. An examination of the DROP SEQ data for the ARC by the Lowell/Tsai lab would indicate that there is some variability in the expression of FAAH in AGRP neurons.

We thank the reviewer for their detailed comment regarding AgRP neuronal expression patterns. It is quite possible that our viral system only reached a subset of AgRP-expressing neurons. Due to the lack of fluorophore in the Agrp-ires-cre mice, we are unable to quantify the number/percent of AgRP neurons that were infected by our viral system. However, we compared our cre-dependent CRISPR/SaCas9 AAV-HA-tag expression in Agrp-ires cre mice to previously published expression profiles in Agrp-ires cre mice using cre-dependent viral delivery systems (1, 2) and see a similar visual expression profile. Nevertheless, we agree with the reviewer that it should be acknowledged that it is possible that only a subset of AgRP neurons are targeted, yet it is still sufficient to recapitulate the exaggerated hyperphagic response to CORT observed in FAAH A/A mice. Furthermore, we should clearly state that it is very possible that other neuronal populations are also involved in mediating anandamide’s downstream effects on glucocorticoid’s orexigenic effects. Therefore, we have added more details in the Results (Section ‘FAAH knockdown exclusively in hypothalamic AgRP neurons is sufficient to mediate the exaggerated hyperphagic response to glucocorticoids and ablate the anorexigenic effects of leptin’) and Discussion. Specifically, within the discussion, we have now added the following:

“Based on the lack of flurophore in AgRP-Ires-cre mice, we could quantify the percent of AgRP neurons targeted by our viral delivery system. Therefore, it is possible that only a subset of AgRP neurons were targeted and lacked FAAH. Nevertheless, this study demonstrates that lowering FAAH expression exclusively in AgRP neurons (targeted or widespread) is sufficient to recapitulate the hyperphagic feeding response of FAAH A/A mice to acute GC treatment. Whether other neuronal populations are likewise involved, remains to be elucidated.”

5) Some mention of the similarities of the mouse and human Faah alleles would be appropriate – in terms of effects on the FAAH steady state concentrations and impact on activity.

We thank the reviewer for pointing out this omission. We agree that we should have described the validity of the FAAH C385A knock-in mouse model in more detail. Therefore, we have added details to the Discussion describing the biochemical similarities between the human and mouse FAAH alleles. Specifically:

“In humans, the common missense mutation (C385A) in FAAH affects FAAH protein stability, with the A/A genotype associated with approximately half the FAAH protein expression and enzymatic activity compared with the C/C genotype (Chiang et al., 2004). More specifically, T-lymphocytes from patients homozygous for the FAAH A/A variant express less than half of the FAAH protein and activity compared to wild-type (WT) lymphocytes. Here we fully characterized the effects of the functional FAAH C385A mutation on GC-dependent metabolic outcomes in a novel C385A knock-in mouse model. Importantly, this FAAH C385A knock-in mouse model recapitulates the reduction in FAAH expression and activity observed in humans expressing the FAAH A/A variant (Dincheva et al., 2015)”

Reviewer #1 (Recommendations for the authors):"The ECS, and specifically anandamide signaling, has an important role in the regulation of body weight."- there is just as much evidence to support 2-AG signaling in the regulation of body weight… maybe more. This should be adjusted.

We thank the reviewer for this comment. We agree that 2-AG is also an important regulator of body weight. Therefore, we have revised the particular sentence mentioned as follows:

“The ECS, and specifically anandamide signaling, has an important role in the regulation of body weight. In particularterms of anandamide, activation of anandamide signaling is known to promote a positive energy balance and weight gain (Mazier et al., 2015).”

Reviewer #2 (Recommendations for the authors):The HA-tag imaging of Cas9 expression does not recapitulate AGRP neuron distribution within the ARC, which is concentrated in the ventromedial aspects of the two ARC nuclei whereas the HA-tag labeling shows a very diffuse and sparse distribution within the dorsolateral aspects of the ARC as well as the spaces between the ARC and the VMN.

The reviewer raises a fair concern. In our discussion, we now discuss our HA-tag expression pattern relative to the typical distribution of AgRP neurons. Briefly, in our study the HA-tagged SaCas9 protein expression spread into the dorsal aspects of the arcuate nuclei whereas AgRP neurons are largely concentrated in the ventral aspects of the arcuate nuclei. This suggests that indeed only a subset of AgRP neurons were targeted. It is possible that we targeted more dorsally positioned AgRP neurons based on our injection strategy wherein we positioned the tip of the cannula towards the dorsal aspects of the arcuate nuclei to avoid penetrating the base of the brain and cerebrospinal fluid. Nevertheless, this study demonstrates that lowering FAAH expression exclusively in AgRP neurons (targeted or widespread) is sufficient to recapitulate the hyperphagic feeding response of FAAH A/A mice to acute GC treatment.